# Intrinsic lipid binding activity of ATG16L1 supports efficient membrane anchoring and autophagy

Leo J Dudley[1,†], Ainara G Cabodevilla[1,†], Agata N Makar[1], Martin Sztacho[2], Tim Michelberger[1], Joseph A Marsh[3], Douglas R Houston[4], Sascha Martens[2], Xuejun Jiang[5] & Noor Gammoh[1,*]

## Abstract

Membrane targeting of autophagy-related complexes is an important step that regulates their activities and prevents their aberrant engagement on non-autophagic membranes. ATG16L1 is a core autophagy protein implicated at distinct phases of autophagosome biogenesis. In this study, we dissected the recruitment of ATG16L1 to the pre-autophagosomal structure (PAS) and showed that it requires sequences within its coiled-coil domain (CCD) dispensable for homodimerisation. Structural and mutational analyses identified conserved residues within the CCD of ATG16L1 that mediate direct binding to phosphoinositides, including phosphatidylinositol 3-phosphate (PI3P). Mutating putative lipid binding residues abrogated the localisation of ATG16L1 to the PAS and inhibited LC3 lipidation. On the other hand, enhancing lipid binding of ATG16L1 by mutating negatively charged residues adjacent to the lipid binding motif also resulted in autophagy inhibition, suggesting that regulated recruitment of ATG16L1 to the PAS is required for its autophagic activity. Overall, our findings indicate that ATG16L1 harbours an intrinsic ability to bind lipids that plays an essential role during LC3 lipidation and autophagosome maturation.

**Keywords** ATG16L1; autophagy; coiled-coil domain; phagophore; PI3P
**Subject Categories** Autophagy & Cell Death; Membrane & Intracellular Transport
**The EMBO Journal (2019) 38: e100554**

## Introduction

While some organelles, such as the endoplasmic reticulum (ER) or mitochondria, are generated by growing and budding from pre-existing organelles, autophagosome formation is initiated through the *de novo* nucleation of membranous structures (Joshi *et al*, 2017). This process requires the activity of distinct protein complexes that act to relay upstream signals in order to facilitate the growth of precursor membranes known as pre-autophagosomal structures (PAS; Lamb *et al*, 2013). Of these protein complexes, the ATG5 complex, comprised of the ATG12~ATG5 conjugate and ATG16L1, plays a pivotal role in both the nucleation of the PAS and the downstream conjugation of the ATG8 ubiquitin-like family of proteins (such as LC3) to phosphatidylethanolamine (PE; Sakoh-Nakatogawa *et al*, 2013). The conjugation of LC3 on the PAS facilitates the maturation of autophagosomes and the recruitment of cargo molecules for their subsequent lysosomal degradation.

In order to better understand how autophagy-related complexes are activated and recruited to the growing PAS, recent studies have started to uncover the genetic and temporal hierarchy of these complexes in mammalian cells (Itakura & Mizushima, 2010; Koyama-Honda *et al*, 2013). Following inhibition of mTORC1 (e.g. by amino acid starvation or small molecule inhibitors), the ULK1 complex is relieved from its inhibitory phosphorylation by mTOR, resulting in its recruitment to the PAS independently of downstream autophagy complexes (Itakura & Mizushima, 2010). The ULK1 kinase can phosphorylate and activate members of the ATG14 complex, containing the class III PI3K kinase Vps34, which facilitates the recruitment of phosphatidylinositol 3-phosphate (PI3P) sensors (such as DFCP1, WIPI1 and WIPI2; Axe *et al*, 2008; Matsunaga *et al*, 2010). FIP200, a component of the ULK1 complex, and WIPI2b can both directly interact with ATG16L1, providing a mechanism for the localisation of the ATG5 complex to the PAS during mTORC1 inactivation (Gammoh *et al*, 2013; Nishimura *et al*, 2013; Dooley *et al*, 2014).

Membrane recruitment of most autophagy complexes is pivotal for their role in autophagy. Ectopic recruitment to the plasma membrane of an ATG5-binding fragment of ATG16L1 results in aberrant and constitutive lipidation of LC3 (Fujita *et al*, 2008). Similar results were also obtained upon plasma membrane tethering of the ATG16L1 binding partner, WIPI2b (Dooley *et al*, 2014). These findings suggest that membrane localisation of ATG16L1 in cells is sufficient to drive the conjugation of LC3 to PE and that its

1   Cancer Research UK Edinburgh Centre, Institute of Genetics and Molecular Medicine, University of Edinburgh, Edinburgh, UK
2   Department of Biochemistry and Cell Biology, Max F. Perutz Laboratories, Vienna Biocenter, University of Vienna, Vienna, Austria
3   Human Genetics Unit, Institute of Genetics and Molecular Medicine, University of Edinburgh, Edinburgh, UK
4   Institute of Quantitative Biology, Biochemistry and Biotechnology, University of Edinburgh, Edinburgh, UK
5   Cell Biology Department, Memorial Sloan Kettering Cancer Centre, New York, NY, USA
   *Corresponding author. Tel: +44 1316518526; E-mail: noor.gammoh@igmm.ed.ac.uk
   †These authors contributed equally to this work

regulation could provide means to fine-tune autophagy. Despite the above proposed hierarchy, it remains unknown how autophagy players are recruited during ULK1-independent autophagy (Gammoh *et al*, 2013). Furthermore, live-cell imaging analyses suggest that both the recruitment and displacement of autophagy proteins to the PAS occurs in an asynchronous manner, indicating that protein–protein interactions are not sufficient to stabilise autophagy complexes on membranes (Koyama-Honda *et al*, 2013). In addition, sequences that correspond to WIPI2b and FIP200 binding sites in mammalian ATG16L1 are absent in yeast Atg16 (Fujioka *et al*, 2010; Gammoh *et al*, 2013; Nishimura *et al*, 2013; Dooley *et al*, 2014). These observations suggest the existence of previously unknown mechanisms that can mediate the localisation of the ATG5 complex to membranes.

Given the central role of the ATG5 complex during various forms of LC3 conjugation, including both canonical and non-canonical autophagy (Fletcher *et al*, 2018), we aimed to investigate how the ATG5 complex is recruited to autophagy-related membranes. In this study, we have identified highly conserved sequences within the coiled-coil domain (CCD) of ATG16L1 that mediate its direct interaction with lipids, thereby enhancing its PAS localisation and autophagic activity.

# Results

## ATG16L1 membrane targeting activity is retained in the absence of ATG5 or WIPI2

To investigate the membrane recruitment of the ATG5 complex, we first addressed the role of either ATG5 or ATG16L1 in PAS targeting. Since deletion of either protein can destabilise the other (Fig 1A, Nishimura *et al*, 2013), we generated knockout cell lines stably expressing GFP-tagged proteins to avoid any discrepancies resulting from reduction in protein levels or ectopic localisation (Li *et al*, 2017). As seen in Fig 1B, the recruitment of stably expressed ATG16L1 to punctate structures was not disrupted in the absence of ATG5. On the other hand, ATG5 showed a diffused pattern of staining in the absence of ATG16L1. Biochemical fractionation further confirmed the finding that ATG16L1 accumulates in membrane fractions in the absence of ATG5 (Fig 1C), as well as in the absence of ATG3 (Gammoh *et al*, 2013). These results suggest a role for ATG16L1 in the membrane targeting of the ATG5 complex. Previous studies show that the PAS localisation of an ATG16L1 mutant lacking both WIPI2b and FIP200 binding (ATG16L1$^{\Delta FBD}$) was markedly reduced but not completely inhibited, suggesting that these interaction partners may act as signalling players that enhance the membrane recruitment of ATG16L1 (Gammoh *et al*, 2013; Nishimura *et al*, 2013). This was further confirmed in WIPI2$^{-/-}$ cells where ATG16L1 puncta formation was reduced but not fully inhibited (Fig 1D and E). Residual ATG16L1-positive puncta formed in WIPI2$^{-/-}$ cells were sensitive to Vps34 inhibition by 3-methyladenine (3′MA) treatment, in agreement with previous data showing the requirement of PI3P for the PAS recruitment of the ATG5 complex (Koyama-Honda *et al*, 2013). Overall, these findings suggest the existence of additional previously undescribed mechanisms that mediate the recruitment of ATG16L1 to the PAS.

## PAS targeting activity of ATG16L1 lies within its CCD

To identify the region within ATG16L1 required for its localisation to the PAS, we examined ATG16L1 puncta formation in U2OS cells expressing a series of ATG16L1 truncation mutants (depicted in Fig 2A). As seen in Fig 2B, the deletion of N-terminal sequences containing the ATG5 binding (fragment Δ1, residues 39–623) or the further downstream linker region (fragment Δ2, residues 120–623) did not affect puncta formation when compared to wild-type ATG16L1 (ATG16L1$^{WT}$). On the other hand, subsequent fragments lacking the CCD of ATG16L1 (fragments Δ3 and Δ4, residues 206–623 and 336–623, respectively) were diffused in cells, suggesting that PAS targeting requires either the dimerisation of ATG16L1 or additional unknown activities within the CCD. To distinguish these two possibilities, we aimed to further delete sequences within the CCD that were predicted to be dispensable for dimerisation. Based on ATG16L1 structural predictions and comparisons to yeast Atg16 (Fujioka *et al*, 2010), we analysed conserved regions within the ATG16L1 CCD that were predicted to not contribute to the dimer-dimer interface. A combined deletion of amino acids 182–205 within the context of the Δ2 fragment (ATG16L1$^{\Delta 2\Delta 182–205}$) resulted in a diffused pattern of staining, indicating the requirement of these residues of ATG16L1 for puncta formation (Fig 2C). We further confirmed that deleting residues 182–205 within the context of the full-length protein (ATG16L1$^{\Delta 182–205}$) did not interfere with the ability of ATG16L1 to interact with FIP200 and ATG5 or homodimerise (Fig 2D and E), whereas a truncation mutant lacking the CCD, but not the WD40 domain, was unable to homodimerise (Fig 2F). When expressed in ATG5$^{-/-}$ cells, the deletion mutant, ATG16L1$^{\Delta 182–205}$, exhibited a diffused pattern of staining (Fig 2G), whereas ATG16L1$^{WT}$ formed punctate structures. These findings confirm that sequences within the CCD are required for PAS targeting but dispensable for previously identified functions of ATG16L1, including dimerisation and binding to FIP200 and ATG5.

## ATG16L1 binds liposomes through CCD sequences

Having shown that the localisation of ATG16L1 to punctate structures requires sequences within its CCD of unknown function, we further investigated the relevance of these sequences in recruiting ATG16L1 to the PAS. Further analyses of the CCD domain region indicated the presence of a hydrophobic region and positively charged residues that could mediate direct lipid binding of ATG16L1 to membranes (Fig 3A). To address this possibility, we used a microscopy-based technique to test the recruitment of rhodamine-labelled small unilamellar vesicles (SUVs) to beads coated with ATG16L1-GFP (Fracchiolla *et al*, 2016), a sensitive approach to detect protein-lipid binding activities. As seen in Fig 3B and C, there was no significant recruitment of liposomes to ATG16L1-GFP-bound beads when using liposome preparations that contained phosphatidylinositol (PI), phosphatidylethanolamine (PE), phosphatidylcholine (PC) and phosphatidylserine (PS) suggesting an inability of ATG16L1 to bind these phospholipids. Consistent with the hypothesis that ATG16L1 can directly bind to autophagy-related membranes, liposome recruitment to ATG16L1-GFP beads was enhanced when incubated with PI3P-containing liposomes, an essential lipid for the biogenesis of autophagosomes (Axe *et al*,

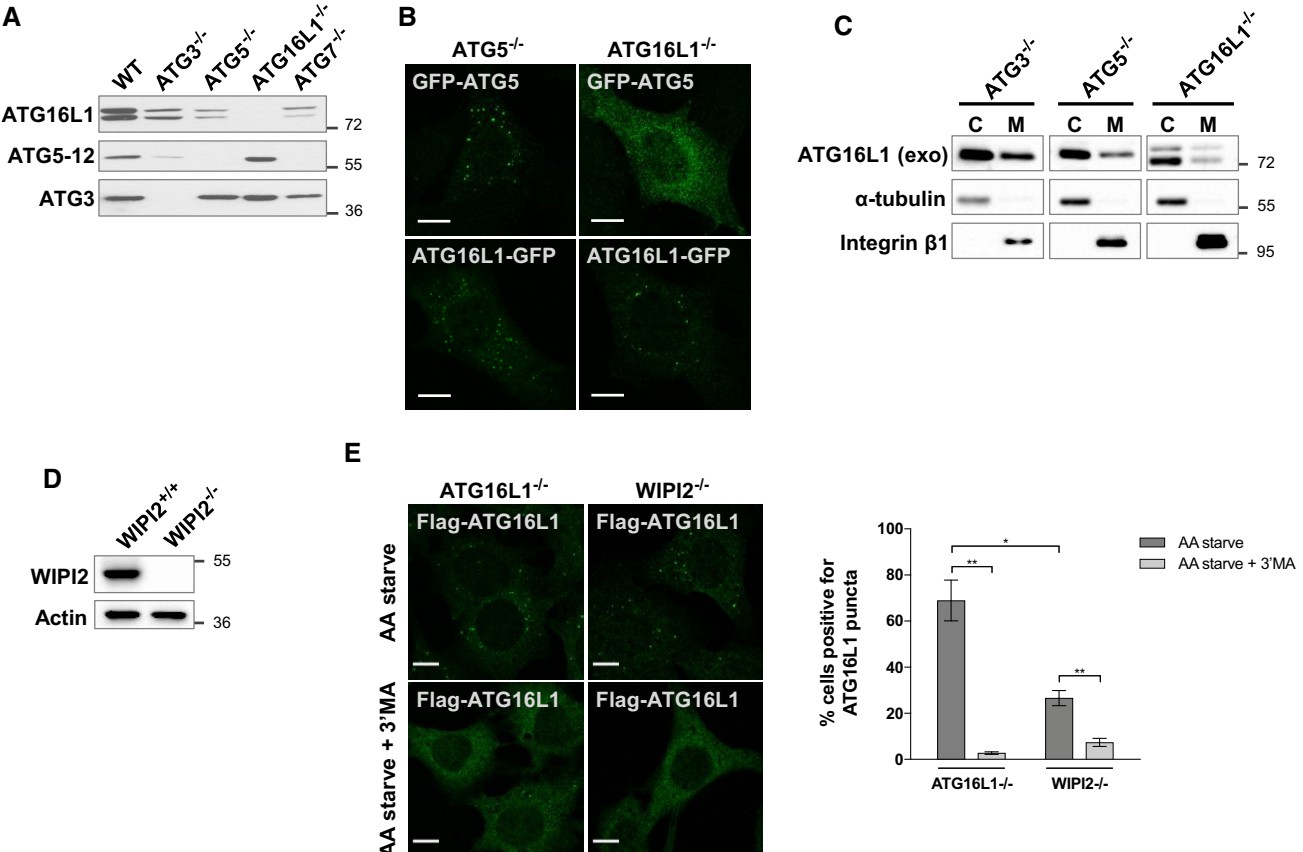

**Figure 1. ATG16L1 membrane targeting activity is retained in the absence of ATG5 or WIPI2.**

A   Analyses of protein expression in lysates of various cell lines by Western blotting against the indicated antibodies.

B   Fluorescence analyses of GFP-ATG5 or ATG16L1-GFP stably expressed in ATG5$^{-/-}$ or ATG16L1$^{-/-}$. Cells were amino acid starved for 2 h prior to fixation and imaging of the GFP fluorescence. Scale bar: 10 μm.

C   Assessment of ATG16L1 levels in the cytosolic (C) and membrane (M) fractions in lysates of the indicated cell lines using Western blot analyses and antibodies against ATG16L1. ATG16L1$^{-/-}$ and ATG5$^{-/-}$ stably expressed ATG16L1-GFP while ATG3$^{-/-}$ stably expressed Flag-S-ATG16L1. Antibodies against α-tubulin and integrin β1 were used as controls for fractionation. Exogenous (exo) ATG16L1 was detected using antibodies against ATG16L1.

D   Lack of WIPI2 expression is confirmed by Western blot analyses in wild-type MEFs (WIPI2$^{+/+}$) and WIPI2$^{-/-}$ cells.

E   Immunofluorescence analyses of ATG16L1$^{-/-}$ and WIPI2$^{-/-}$ stably expressing Flag-S-ATG16L1. Cells were amino acid starved (AA starve) for 2 h in the presence or absence of 3'MA (and additional pretreatment for 30 min) followed by fixation and immunostaining using antibodies against Flag tag to detect ATG16L1. Scale bar: 9 μm. Right panel represents quantification of three independent experiments and error bars depicting SEM values. *$P \leq 0.05$, **$P \leq 0.01$ (pairwise unpaired Student's *t*-test).

2008). Moreover, liposome preparations containing phosphatidylinositol 4-phosphate (PI4P) or phosphatidylinositol 4,5-bisphosphate (PI(4,5)P2) also resulted in enhanced recruitment to ATG16L1-GFP containing beads, indicating that these membrane phospholipids can also bind ATG16L1. The binding of ATG16L1 to liposomes was independent of ATG5 or its WD40 domain as a deletion mutant of ATG16L1 lacking its C-terminal half was able to bind liposomes when purified from ATG5$^{-/-}$ cells (Fig EV1A and B). We further confirmed the ability of wild-type ATG16L1 to bind PI3P in an independent assay using lipid-coated beads, where we also detected its binding to phosphatidylinositol 3,4-bisphosphate (PI(3,4)P2) (Fig EV1C). In contrast, ATG16L1 did not significantly bind to phosphatidylinositol 3,4,5-trisphosphate (PI(3,4,5)P3) when compared to control beads, suggesting a degree of specificity for phosphoinositides. Interestingly, this assay also revealed that ATG16L1 has low binding affinity for PE and PS, while also exhibiting an affinity for

PA, although this was found to be non-significant compared to the control beads.

The ability of ATG16L1 to bind PI3P suggests that positively charged residues, potentially located within or juxtaposed to amino acids 182–205 of its CCD, may mediate this interaction. Although previous studies have identified interactions between coiled-coil domains and phospholipids (Horikoshi *et al*, 2011; Zheng *et al*, 2014), the structural bases underlying these have not yet been directly elucidated. To explore the potential structural mechanism of ATG16L1-lipid interaction, we performed structural prediction analyses of a short region of mouse ATG16L1 CCD and modelled its interaction with PI3P embedded in a lipid bilayer. These analyses predict that ATG16L1 CCD could potentially interact with the negatively charged headgroup of PI3P by lying flat on the membrane surface (Fig 3D). Molecular dynamics simulation of the protein in association with PI3P in a model lipid bilayer indicated that this

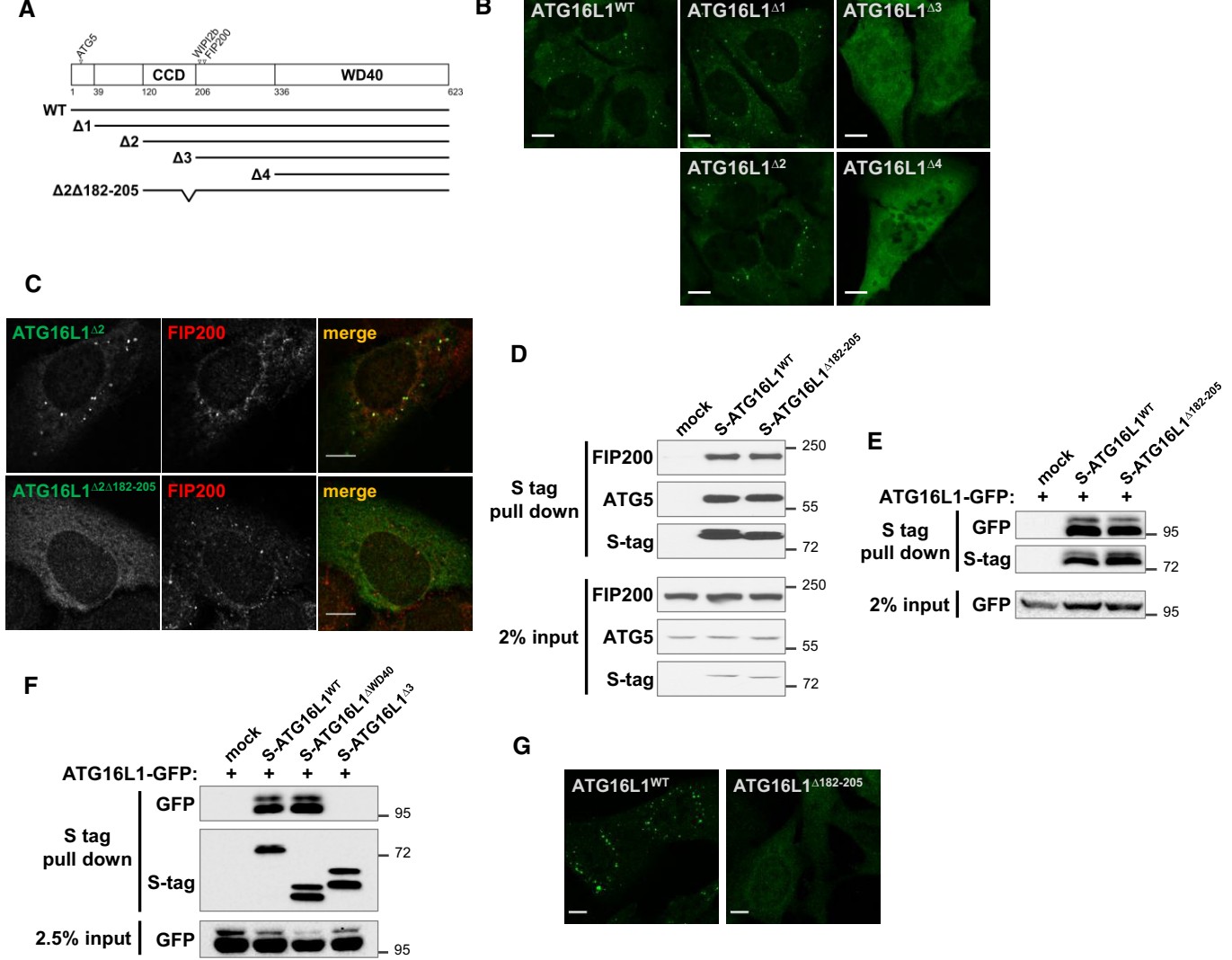

**Figure 2. PAS targeting activity of ATG16L1 lies within its CCD.**

A  Schematic presentation of ATG16L1 fragments and mutants used in this study. The following fragments encompassed the indicated amino acids: wild type (WT): 1–623; Δ1: 39–632; Δ2: 120–623; Δ3: 206–623 and Δ4: 336–623. Mutant Δ2Δ182–205 consists of the Δ2 fragment with an additional deletion in amino acids 182–205. All constructs contained a Flag-S tag at the N-terminal end.

B  Fragments depicted in (A) were expressed in U2OS cells and amino acid starved for 5 h followed by fixation and immunostaining against S tag to detect S-ATG16L1. Scale bar: 10 μm.

C  U2OS cells expressing Flag-ATG16L1$^{Δ2}$ or Flag-ATG16L1$^{Δ2Δ182–205}$ were treated as in (B) and stained using antibodies against Flag tag (to detect ATG16L1, green) and FIP200 (red). Scale bar: 10 μm.

D  Protein–protein interaction assay in 293T cells transiently transfected with the indicated Flag-S-tagged ATG16L1 constructs. S tag pull-down was performed and protein complexes were analysed by immunoblotting using the indicated antibodies.

E  Homodimerisation assay in 293T cells transiently transfected with ATG16L1-GFP and the indicated Flag-S-tagged ATG16L1 constructs. S tag pull-down was performed and protein complexes were analysed by immunoblotting using the indicated antibodies.

F  Homodimerisation assay similar to (E).

G  Flag-ATG16L1$^{WT}$ or Flag-ATG16L1$^{Δ182–205}$ were stably expressed in ATG5$^{−/−}$ cells and analysed by immunofluorescence using antibodies against Flag tag to detect ATG16L1. Scale bar: 9 μm.

association was stable over the course of the trajectory. Interestingly, helices and coiled-coil domains have also been previously shown to interact with lipid bilayers by lying flat on the membrane surface, although adopting a different mechanism than that predicted for ATG16L1 (utilising primarily hydrophobic rather than electrostatic interactions; Pluhackova *et al*, 2015; Woo & Lee, 2016).

Our homology modelling highlights three residues, including K179, R193 and the further upstream residue I171, which line the outer faces of the coiled-coil and are solvent-exposed, thereby free to interact with the phosphate groups of PI3P or PI(3,4)P2. These residues are conserved in yeast Atg16 but are missing from ATG16L2, a protein closely related to ATG16L1 that does not localise to the PAS

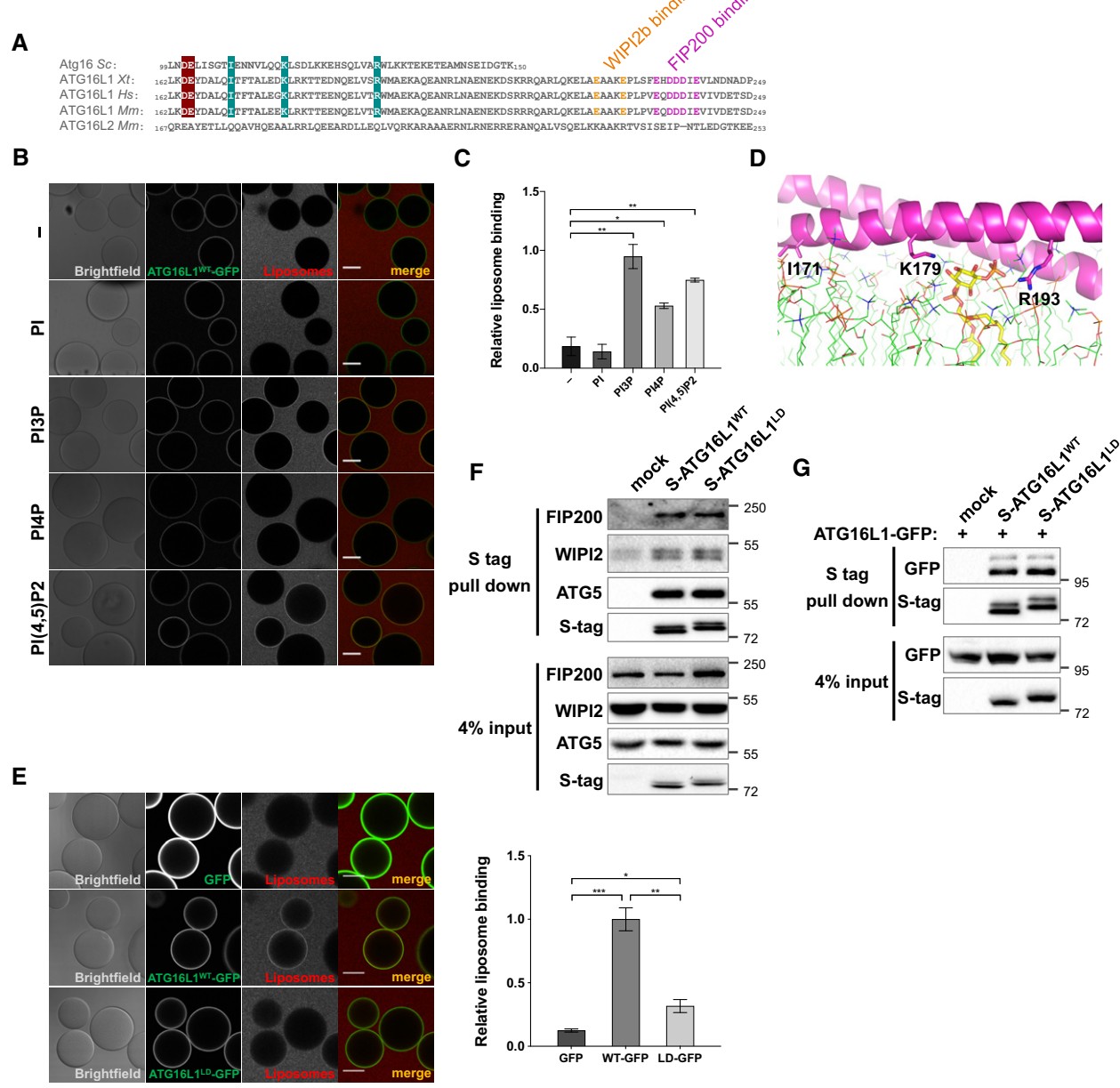

**Figure 3. ATG16L1 binds liposomes through CCD sequences.**

A   Sequence alignment of ATG16L1 CCD segment from various species and ATG16L2 (*Sc*: *Saccharomyces cerevisiae*; *Xt*: *Xenopus (Silurana) tropicalis*; *Hs*: *Homo sapiens*; *Mm*: *Mus musculus*). Residues that mediate WIPI2b and FIP200 binding are highlighted in orange and magenta, respectively. Cyan-shaded residues (I171, K179 and R193) are exposed conserved residues predicted to not contribute to the dimer-dimer interface and are mutated in this study. Cherry-shaded residues are mutated in Figs 6 and 7.

B   Microscopy-based protein–liposome binding assay. ATG16L1-GFP immobilised on beads incubated in the presence of rhodamine-labelled liposome preparations containing the indicated phosphoinositides. Scale bar: 50 µm.

C   Quantification of relative liposome binding in (B).

D   Structural modelling of ATG16L1 residues 160–205 (magenta helix) in the presence of lipid bilayer (green lines). Highlighted residues include I171, K179 and R193 as sticks, which are mutated in this study. A PI3P molecule is shown as a yellow stick embedded in the lipid bilayer and interacting with the highlighted positively charged residues of ATG16L1.

E   Microscopy-based protein–liposome binding assay as in (B). ATG16L1^WT- and ATG16L1^LD-GFP immobilised on beads were incubated with rhodamine-labelled, PI3P-positive liposome preparations. Scale bar: 50 µm. Right panel shows quantification of liposome binding relative to ATG16L1^WT from three independent experiments including SEM values.

F   Protein–protein interaction assay in 293T cells transiently transfected with the indicated S-tagged ATG16L1 constructs. S tag pull-down was performed and protein complexes were analysed by immunoblotting using the indicated antibodies.

G   Dimerisation assay in 293T cells transiently transfected with ATG16L1-GFP and the indicated S-tagged ATG16L1 constructs and analysed as in (F).

Data information: Quantifications depict means and error bars (SEM) from at least three independent experiments. *$P \leq 0.05$, **$P \leq 0.01$, ***$P \leq 0.001$ (pairwise unpaired Student's *t*-test).

despite its ability to bind ATG5 and homodimerise (Ishibashi et al, 2011). As predicted, mutation of these three residues in ATG16L1 to aspartic acid (I171D, K179D and R193D) strongly reduced the recruitment of liposomes to ATG16L1-coated beads (Fig 3E). Hence, we named this mutant ATG16L1$^{LD}$ for lipid binding-deficient ATG16L1 mutant. Given that these residues are adjacent to WIPI2b and FIP200 binding sites, we further confirmed that ATG16L1$^{LD}$ did not interfere with the ability of ATG16L1 to bind these two interactors nor interfere with ATG16L1 homodimer formation (Fig 3F and G). Overall, these data suggest that ATG16L1 contains residues within its CCD that mediate interactions to negatively charged lipids but are dispensable for its dimerisation and binding to WIPI2 and FIP200.

### Binding of ATG16L1 to lipids is required for PAS recruitment

To dissect the functional relevance of ATG16L1 lipid binding in cells, we further analysed the localisation of ATG16L1$^{WT}$ and ATG16L1$^{LD}$, stably expressed in ATG16L1$^{-/-}$ cells. When compared to ATG16L1$^{WT}$, we observed a strong inhibition of ATG16L1$^{LD}$ recruitment to puncta positive for ATG5 and WIPI2 during amino acid starvation (Fig 4A and B), suggesting that ATG16L1 lipid binding is required for its efficient recruitment to the PAS. Additionally, the overall intensity of ATG16L1$^{LD}$ puncta was significantly reduced compared to that of ATG16L1$^{WT}$, implying that ATG16L1 is recruited less efficiently to these sites (Fig 4C). Furthermore, enhanced formation of WIPI2-positive puncta, as observed by reconstituting ATG16L1$^{-/-}$ cells with ATG16L1$^{WT}$, was impaired in ATG16L1$^{LD}$-expressing cells, indicating that ATG16L1 lipid binding may influence early autophagic events (Fig 4B). Residual ATG16L1$^{LD}$ punctate structures could potentially be due to its ability to bind upstream autophagy players, including WIPI2b and FIP200, which have been proposed to facilitate its recruitment to the PAS (Gammoh et al, 2013; Nishimura et al, 2013; Dooley et al, 2014). To test this possibility, we stably expressed ATG16L1$^{LD}$ in WIPI2$^{-/-}$ cells and observed a further reduction in puncta formation of this mutant to levels comparable to background levels in mock infected cells (Fig 4D). Overall, these data suggest that the efficient recruitment of ATG16L1 to the PAS relies on its ability to interact with both PI3P through CCD sequences and protein binding partners, such as WIPI2b.

### Binding of ATG16L1 to lipids is required for autophagy

To examine the functional impact of disrupted PAS localisation in the lipid binding-deficient mutant, ATG16L1$^{LD}$, we further assessed its ability to mediate LC3 lipidation in cells. When compared to ATG16L1$^{-/-}$ cells reconstituted with ATG16L1$^{WT}$, ATG16L1$^{LD}$-expressing cells exhibited a strong inhibition of LC3 lipidation during amino acid starvation (Fig 5A). To confirm that autophagy was inhibited, we measured the levels of p62, an adaptor protein that is degraded during autophagy, following amino acid starvation. Consistent with the LC3 lipidation results, p62 degradation was impaired in ATG16L1$^{LD}$-expressing cells, suggesting that autophagic flux was also inhibited (Fig 5A). Furthermore, we tested LC3 lipidation induced by carbonyl cyanide m-chlorophenylhydrazone (CCCP, 10 μM) treatment, shown to induce the selective degradation of mitochondria (Narendra et al, 2008), and similarly observed

defective LC3 lipidation in ATG16L1$^{LD}$-expressing cells (Fig 5B). Similar results were obtained during autophagy induced by glucose starvation, suggesting that the lipid binding domain of ATG16L1 is also required for autophagy induced in the absence of mTORC1 inhibition and ULK1-complex activation (Fig 5C). Given that non-canonical LC3 lipidation can occur on single membranes and requires the ATG5 complex but not additional upstream autophagy machinery, such as WIPI2 or the ULK1 complex, we further examined whether ATG16L1$^{LD}$ could support LC3 lipidation during treatment with monensin, ammonium chloride (NH$_4$Cl) or CCCP (100 μM), known to act as ionophores and/or lysosomotropic agents (Jacquin et al, 2017). Consistent with previous results, these treatments required the activity of ATG16L1$^{WT}$ to support LC3 lipidation (Fig 5D and E; Fletcher et al, 2018), whereas LC3 lipidation was strongly diminished in ATG16L1$^{LD}$-expressing cells. Collectively, these data suggest that the lipid binding activity of ATG16L1 is required to facilitate both LC3 lipidation induced by various stimuli and the efficient degradation of autophagic cargo.

In addition to binding FIP200 and WIPI2b, ATG16L1 has also been shown to bind Rab33B through sequences within its middle region (Itoh et al, 2008). The relevance of Rab33B binding remains to be further explored as published studies suggest that shRNA-mediated inhibition of Rab33B expression did not affect LC3 lipidation while overexpression of a GTP-hydrolysis-deficient (constitutively active) mutant of Rab33B enhanced LC3 lipidation (Itoh et al, 2008, 2011). We further tested whether mutating lipid binding residues within the CCD of ATG16L1 affected the interaction between ATG16L1 and Rab33B. Surprisingly, ATG16L1$^{LD}$ exhibited a strongly diminished affinity to GFP-Rab33B compared to ATG16L1$^{WT}$ (Fig EV2A). To assess whether genetic inhibition of Rab33B influenced autophagy in a manner that mimicked mutating the residues involved in ATG16L1 lipid binding, we generated Rab33B$^{-/-}$ cells using CRISPR/Cas9-mediated gene editing in MEF cells. Rab33B$^{-/-}$ cells did not exhibit reduced LC3 lipidation (Fig EV2B) or disrupt ATG16L1 puncta formation (Fig EV2C) in a manner that resembled ATG16L1$^{LD}$-reconstituted cells (Figs 4A and 5A). Altogether, these data suggest that the functional relevance of ATG16L1 binding to Rab33B is distinct from its binding to PI3P (Ishibashi et al, 2011; Fujita et al, 2013).

### Negatively charged residues within the CCD of ATG16L1 weakens its interaction to lipids

The above studies indicate that the lipid binding ability of ATG16L1 is required for its membrane recruitment and subsequent LC3 lipidation. Because the ATG5 complex, along with upstream autophagy complexes, are only transiently recruited to the autophagosome (Karanasios et al, 2013; Koyama-Honda et al, 2013), we aimed to examine the implication of persistent localisation of ATG16L1 to the PAS. Our structural analyses predicted that multiple negatively charged residues that are exposed on the same heptad repeat positions as residues I171, K179 and R193 (Fig 6A) are likely to be involved in the binding of the CCD to lipids on the membrane surface due to their close proximity to the lipid bilayer. Mutating these negatively charged residues (namely D164, E165, E178, E185 or E186) is likely to enhance the affinity of ATG16L1 for lipids. To test this, we mutated two conserved adjacent residues, D164 and E165 (highlighted in Fig 3A), to alanine (D164A

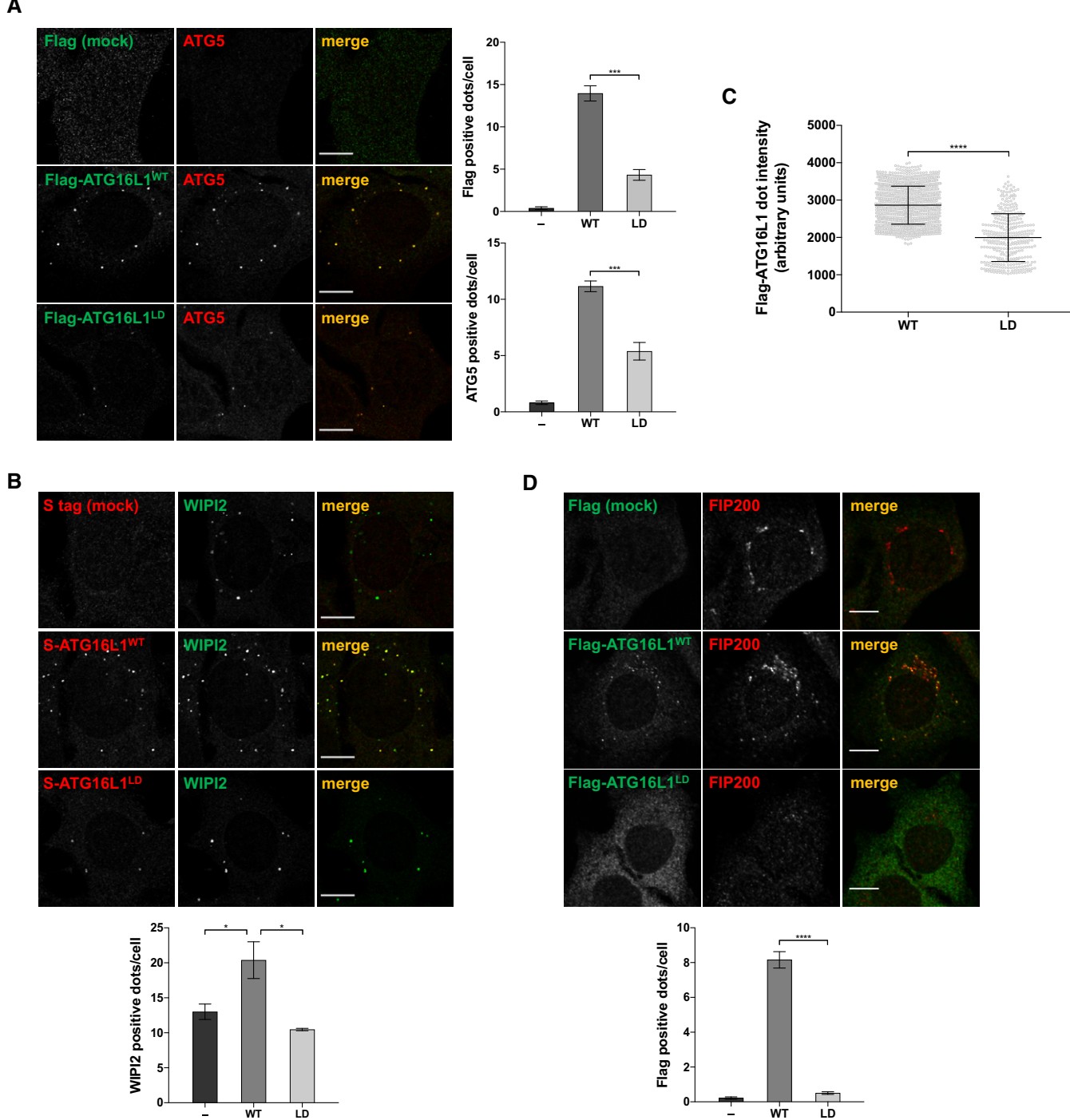

**Figure 4. Binding of ATG16L1 to lipids is required for PAS recruitment.**

A   ATG16L1$^{-/-}$ stably expressing the indicated Flag-S-ATG16L1 constructs were amino acid starved for 2 h prior to immunofluorescence analyses using antibodies against Flag tag (green) and ATG5 (red). Scale bar: 9 μm. Right panels show quantifications of average number of Flag- and ATG5-positive dots per cell.

B   Cells, as in (A), were immunostained using antibodies against S tag (to detect ATG16L1, red) and WIPI2 (green). Scale bar: 9 μm. Lower panel shows quantification of average number of WIPI2-positive dots per cell.

C   Average intensities of individual Flag-ATG16L1 dots in (A). Underlying grey circles represent individual data points. Dots were quantified ($n$ = 1,225 for ATG16L1$^{WT}$; $n$ = 334 for ATG16L1$^{LD}$).

D   WIPI2$^{-/-}$ cells reconstituted with the indicated Flag-S-ATG16L1 constructs were amino acid starved for 2 h prior to immunofluorescence analyses using antibodies against Flag tag (green) and FIP200 (red). Scale bar: 9 μm. Lower panel shows quantification of average number of Flag-positive dots per cell.

Data information: Quantifications depict means and error bars (SEM) from at least three independent experiments. *$P ≤ 0.05$, ***$P ≤ 0.001$, ****$P ≤ 0.0001$ (pairwise unpaired Student's $t$-test).

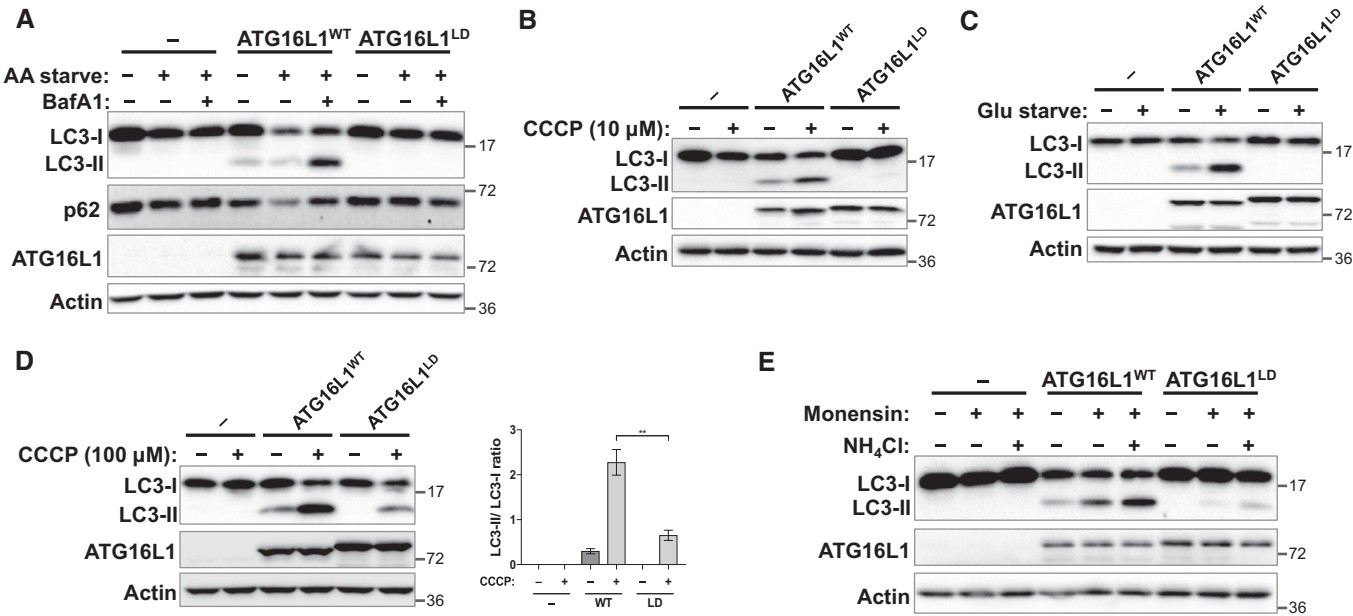

**Figure 5. Binding of ATG16L1 to lipids is required for autophagy.**

ATG16L1$^{-/-}$ cells were reconstituted with ATG16L1$^{WT}$- or ATG16L1$^{LD}$-expression constructs and autophagy assessed during treatment with various stimuli followed by immunoblotting using the indicated antibodies.

A  Cells were amino acid starved (AA starve) to induce mTORC1-dependent autophagy in the presence or absence of BafA1 for 2 h prior to lysis.
B  Mitophagy was induced by CCCP (10 μM) treatment for 6 h prior to lysis.
C  Glucose starvation (Glu starve) for 20 h was used to induce mTORC1-independent autophagy. BafA1 was added to all conditions 2 h prior to lysis.
D  LC3-associated phagocytosis (LAP)-like LC3 lipidation was induced by treating cells with CCCP (100 μM) for 2 h prior to lysis. Right panel shows quantification of LC3-II/LC3-I ratio. Means and error bars (SEM) are shown (*n* = 4). **$P \leq 0.01$ (pairwise unpaired Student's *t*-test).
E  LAP-like LC3 lipidation induced by monensin or ammonium chloride (NH₄Cl) for 2 h prior to lysis.

and E165A) and termed the resulting mutant ATG16L1$^{LE}$ for its potentially lipid binding-enhancing ability. Furthermore, mutating residues D164 and E165 retained the ability of ATG16L1 to bind Rab33B (Fig EV3A). To test lipid binding activity, we generated recombinant ATG16L1 protein spanning residues 72–307, which lacks both the ATG5 binding domain and the C-terminal WD40 domain, in order to enhance protein expression as shown previously (Fig 5B, Archna & Scrima, 2017). We next tested the lipid binding activity of the ATG16L1$^{LE}$ mutant using liposome co-sedimentation assays and SUVs prepared by sonication. Similar to previously published data (Romanov *et al*, 2012), we observed a weak interaction between ATG16L1$^{WT}$ and liposomes under these conditions (Fig 6C). Importantly, this fragment of ATG16L1 (ATG16L1$_{72–307}^{WT}$) was able to bind liposomes in the imaging-based experiment, suggesting that its weak pelleting in the liposome co-sedimentation assay is not due to the lack of ATG5 binding or WD40 domains but due to differences in the experimental approach (Fig EV3B). Interestingly, ATG16L1$^{LE}$ exhibited enhanced liposome binding, suggesting that residues D164 and E165 can modulate the affinity of ATG16L1 for lipids (Fig 6C). Similar to ATG16L1$^{WT}$ protein tested using the GFP-coated beads setting (Fig 3B), the enhanced liposome binding ability of ATG16L1$^{LE}$ was also dependent on PI3P (Fig 6D). Importantly, mutations in residues D164 and E165 and lipid binding residues (I171, K179 and R193) in the combined mutant ATG16L1$^{LELD}$ (D164A, E165A, I171D, R179D and K193D) abolished the lipid binding activity of ATG16L1 (Fig 6C), suggesting that ATG16L1$^{LE}$ enhances the lipid

binding activity of CCD residues. In the context of our model (Fig 3D), we conclude that the loss of negative charges on the exposed surface of this CCD region enhances the affinity of ATG16L1 for negatively charged lipid headgroups, such as PI3P.

**Persistent localisation of ATG16L1 to PAS inhibits autophagy**

To assess the consequence of persistent lipid binding in cells, we tested the recruitment of ATG16L1$^{LE}$ to the PAS. When ATG16L1$^{WT}$ and ATG16L1$^{LE}$ were expressed in ATG16L1$^{-/-}$ cells, similar numbers of cells were positive for ATG16L1 puncta following amino acid starvation (Fig 7A and C). Furthermore, these puncta showed comparable colocalisation with WIPI2 (Fig 7A). Importantly, puncta formed by ATG16L1$^{LE}$ were sensitive to Vps34 inhibition by 3′MA treatment suggesting the relevance of PI3P in its PAS localisation (Fig 7B and C). Interestingly, ATG16L1$^{LE}$ localised to PAS structures that were positive for WIPI2 and responsive to Vps34 inhibition under unstarved conditions or upon replenishment with full growth media following amino acid starvation (Figs 7D and EV4A–C). Importantly, these observations indicate that ATG16L1$^{LE}$ persistently localises to autophagic structures. To test whether persistent localisation of ATG16L1 to the PAS resulted in enhanced or deleterious effects on autophagy, we tested LC3 lipidation in ATG16L1$^{-/-}$ cells expressing either ATG16L1$^{WT}$ or ATG16L1$^{LE}$. As can be seen in Fig 7E, LC3 lipidation was significantly inhibited in ATG16L1$^{LE}$-expressing cells. Punctate structures formed by ATG16L1$^{LE}$ under unstarved conditions could be due to

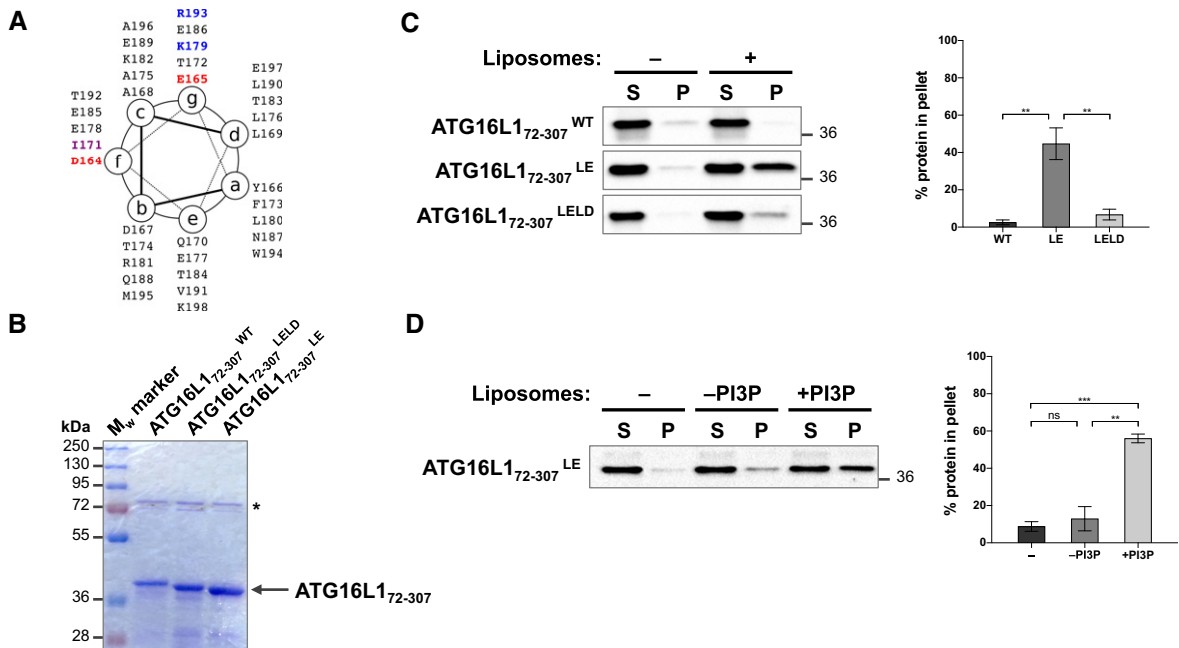

**Figure 6. Negatively charged residues within the CCD of ATG16L1 weakens its interaction to lipids.**

A   ATG16L1 CCD region 164–198 depicted on a heptad repeat with the relevant residues highlighted in red (acidic; D164 and E165), purple (hydrophobic; I171) and blue (basic; K179 and R193).

B   Coomassie gel of recombinant ATG16L1$_{72-307}$ wild type and mutants. * indicates bacterial protein.

C   Liposome co-sedimentation assay using recombinant ATG16L1 protein and sonicated liposomes. Supernatant (S) and pellet (P) were analysed by immunoblotting against T7 tag to detect ATG16L1. Quantification of percentage protein in the pellet fraction was calculated from three independent experiments (right panel). Error bars (SEM). **$P \leq 0.01$ (unpaired Student's $t$-test).

D   Liposome co-sedimentation assay, as in (C), using ATG16L1$_{72-307}$$^{LE}$ and sonicated liposomes that contain or lack PI3P. Quantification of percentage protein in the pellet fraction was calculated from three independent experiments (right panel) with error bars depicting SEM values. ns $P > 0.05$, **$P \leq 0.01$, ***$P \leq 0.001$ (pairwise unpaired Student's $t$-test).

inhibited phagophore maturation which has been shown to accumulate the ATG5 complex on the PAS (Sou *et al*, 2008; Gammoh *et al*, 2013). To test whether ATG16L1$^{LE}$ affects autophagy in cells that contain endogenous ATG16L1, we expressed ATG16L1 constructs in WIPI2$^{-/-}$ cells where autophagic vesicle formation is impaired but not fully inhibited (Bakula *et al*, 2017). Expression of ATG16L1$^{LE}$, but not ATG16L1$^{WT}$, resulted in inhibited LC3 lipidation in a dominant negative manner, thereby suggesting that enhanced lipid binding can have deleterious effects on phagophore maturation during both basal and amino acid starvation-induced autophagy (Fig 7F). We did not observe similar inhibition of LC3 lipidation upon ATG16L1$^{LE}$ expression in wild-type MEFs (data not shown). Since wild-type ATG16L1 puncta formation was markedly reduced in WIPI2$^{-/-}$ cells (Fig 1E), the dominant negative effects of ATG16L1$^{LE}$ are likely to require higher expression levels in the context of wild-type cells in order to compete with endogenous ATG16L1$^{WT}$. Overall, these results suggest that regulated localisation of ATG16L1 to the PAS is required for autophagy.

### Binding of ATG16L1 to lipids is required for ferroptosis

Ferroptosis is a recently described cell death mechanism dependent on iron availability (Gao *et al*, 2015) and the induction of autophagy (Gao *et al*, 2016). Metabolic stress induced by amino acid starvation

in the presence of serum has been shown to induce ferroptosis which can be suppressed by the genetic inhibition of autophagy or use of dialysed serum to deplete glutamine. To test whether the lipid binding mutants of ATG16L1 can mediate the role of autophagy in supporting ferroptosis, we cultured cells in amino acid free media in the presence of 10% FBS (to induce ferroptosis) or 10% dialysed FBS (diFBS, as a control). As previously shown, cell death measured by propidium iodide (PI) staining was robustly induced in ATG16L1$^{-/-}$ cells stably expressing ATG16L1$^{WT}$ compared to parental ATG16L1$^{-/-}$ cells in conditions that induce ferroptosis (Figs 7G, and EV4D and E). Importantly, cell death was strongly inhibited in cells expressing ATG16L1$^{LD}$ or ATG16L1$^{LE}$, emphasising the important role of regulated ATG16L1-lipid binding for autophagy and its functional activities.

## Discussion

Our study shows for the first time that ATG16L1 harbours an intrinsic ability to bind autophagy-related membranes through direct interaction with PI3P. This interaction requires conserved sequences within the CCD of ATG16L1 that, when mutated, result in inhibited recruitment to the PAS and LC3 lipidation. Importantly, enhancing lipid binding of ATG16L1 by mutating negatively

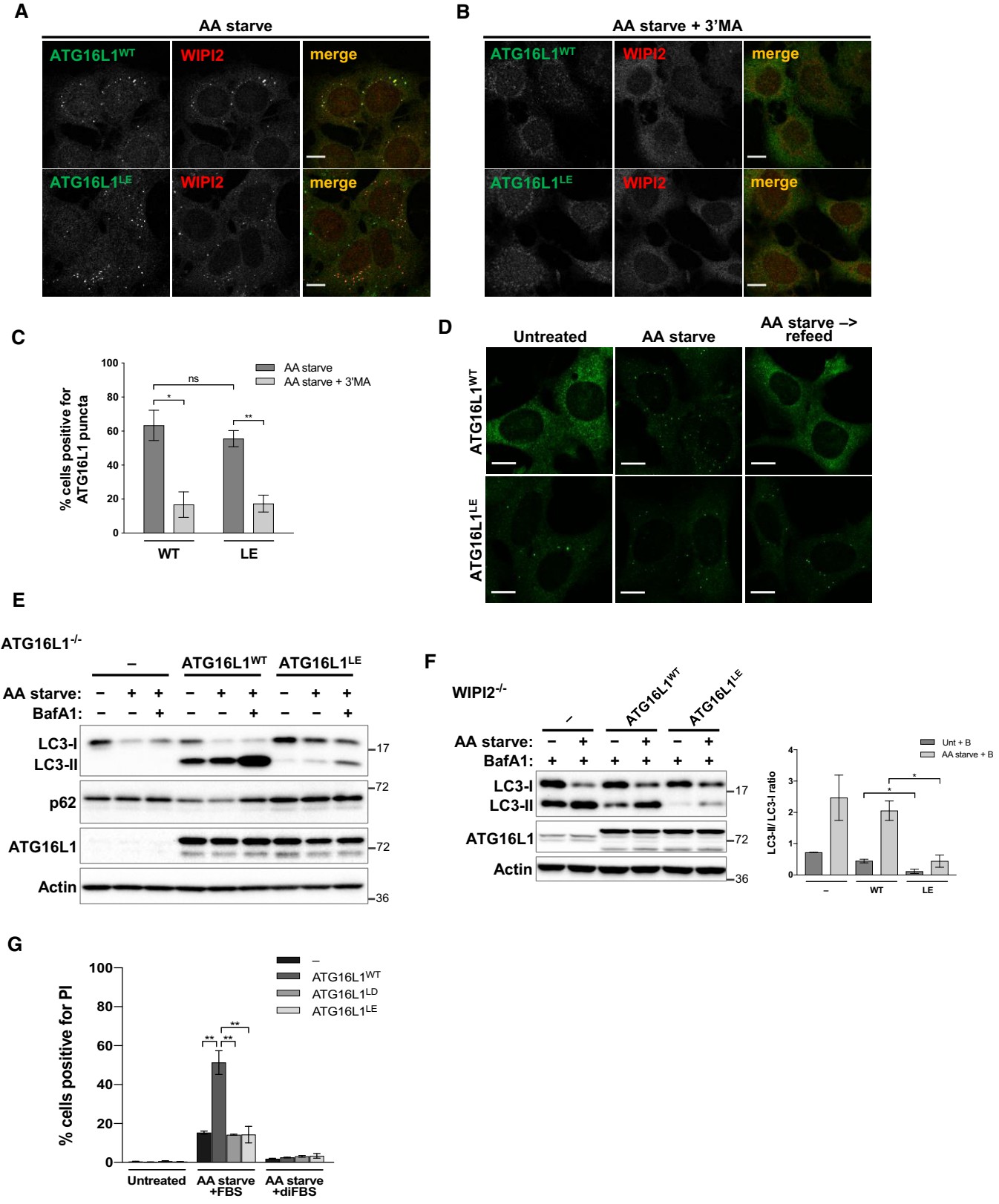

Figure 7.

**Figure 7. Persistent localisation of ATG16L1 to PAS inhibits autophagy.**

ATG16L1$^{-/-}$ cells were reconstituted with ATG16L1$^{WT}$- or ATG16L1$^{LE}$-expression constructs and autophagy assessed during amino acid starvation (AA starve) followed by immunoblotting or immunofluorescence analyses.

A  Immunofluorescence analyses of amino acid starved cells (2 h) using antibodies against WIPI2 (red) or ATG16L1 (green). Scale bar: 9 μm.

B  Cells as in (A) with the addition of 3′MA (and pretreatment with the drug for 30 min). Scale bar: 9 μm.

C  Quantification of percentage of cells positive for ATG16L1 puncta in (A) and (B).

D  Cells were left untreated, amino acid starved for 2 h or amino acid starved for 2 h followed by refeeding with full growth media for 1 h. Puncta formation was assessed by immunofluorescence staining using antibodies against S tag to detect ATG16L1. Scale bar: 10 μm.

E  Immunoblot analyses of amino acid starved cells in the presence or absence of BafA1 for 2 h. Cell lysates were analysed using the indicated antibodies.

F  WIPI2$^{-/-}$ cells reconstituted with the indicated ATG16L1 constructs were amino acid starved for 2 h. BafA1 treatment was included in all conditions for 2 h prior to lysis. Cell lysates were analysed by immunoblotting using the indicated antibodies. Quantification of LC3-II/LC3-I ratio of three independent experiments is shown on the right panel.

G  Ferroptosis assay in ATG16L1$^{-/-}$ cells stably expressing ATG16L1$^{WT}$, ATG16L1$^{LD}$ or ATG16L1$^{LE}$. Cells were cultured in amino acid free media in the presence of 10% FBS or 10% dialysed FBS (diFBS) for 24 h. Quantification of percentage of PI-positive cells from at least three independent experiments is shown.

Data information: Quantifications depict means and error bars (SEM) from at least three independent experiments. ns $P > 0.05$, *$P \leq 0.05$, **$P \leq 0.01$ (pairwise unpaired Student's $t$-test).

charged residues also disrupted LC3 lipidation, suggesting that the tightly regulated recruitment of ATG16L1 to the PAS is important to permit the proper maturation of the phagophore. This is reflected by the weak lipid binding activity of ATG16L1 in the liposome co-sedimentation assay, which indicates that additional factors, such as interactions with other autophagy complexes and/ or post-translational modifications on ATG16L1, may enhance its recruitment to the PAS in cells.

Given that inhibiting the catalysis of PI3P impairs autophagy (Blommaart *et al*, 1997), the PI3P dependency of ATG16L1 lipid binding implies its role in directly binding to sites of autophagosome biogenesis. It remains unclear whether the affinity of ATG16L1 for PI3P or other phosphoinositide species provides a mechanism for its targeting to additional membrane compartments including the plasma membrane and endocytic vesicles (Ravikumar *et al*, 2010; Puri *et al*, 2013). Interestingly, a recent study shows that the recruitment of WIPI2- to PI3P-positive recycling endosomes requires both its lipid and protein interactions with PI3P and RAB11A, respectively (Puri *et al*, 2018). In a parallel mechanism, the recruitment of ATG16L1 to the PAS, as well as other membrane compartments, could require both lipid and protein interactions. Furthermore, ATG16L1-dependent lipidation of LC3 on single membranes, for example during treatment with ionophores and lysosomotropic agents, has been shown to occur independently of FIP200, WIPI2 and PI3P (Florey *et al*, 2015; Fletcher *et al*, 2018). The ability of ATG16L1 to bind other phosphoinositides as well as potentially unidentified protein interactors may be required to mediate the role of ATG16L1 during non-canonical LC3 lipidation. Additionally, potential binding of ATG16L1 to other phospholipids, such as PA, may be required for unknown functions of ATG16L1 that are independent of PAS recruitment.

The middle region of ATG16L1 has also been shown to bind Rab33B with the specific residues that mediate this interaction not previously identified (Itoh *et al*, 2008; Ishibashi *et al*, 2011). Intriguingly, we find that mutating residues I171D, K179D and R193D in the lipid-deficient mutant, ATG16L1$^{LD}$, strongly reduced its binding to Rab33B. It is therefore possible that the proper recruitment of ATG16L1 to the PAS, mediated through its lipid binding, is required for Rab33B binding. It is also possible that the binding of the CCD of ATG16L1 to lipids and Rab33B occur in a mutually exclusive manner, thereby regulating the function of ATG16L1 during various stages of autophagy. Indeed, previous work has shown that a

RabGAP for Rab33B, OATL1, can influence the autophagosome–lysosome fusion, thereby suggesting that Rab33B may also act at later stages of autophagosome maturation (Itoh *et al*, 2011). Future studies addressing these questions will help further understand the regulation of ATG16L1 activities.

There are discrepancies regarding the proposed hierarchies in which autophagy players are recruited to the growing autophagosome (Koyama-Honda *et al*, 2013; Dooley *et al*, 2014). The recruitment of these complexes could occur independently of each other given the individual intrinsic membrane-binding abilities of the ULK1 complex (Karanasios *et al*, 2013), WIPI proteins (Baskaran *et al*, 2012), and ATG5 complex (this study). However, the efficiency of their recruitment appears to be interdependent. For instance, ATG16L1 and WIPI2 puncta are reduced in WIPI2$^{-/-}$ and ATG16L1$^{-/-}$ cells, respectively (this study and Dooley *et al*, 2014). Furthermore, multiple ATG-related proteins can recognise PI3P at the phagophore, many of which harbour non-canonical PI3P binding motifs (including WIPI2b and ATG16L1). The question remains as to whether these PI3P effectors can simultaneously bind to PI3P or whether their binding occurs consecutively. If the latter is true, this may represent a mechanism whereby the ATGs can regulate the recruitment and displacement of one another.

Consistent with our findings using the lipid binding-enhanced mutant ATG16L1$^{LE}$, membrane tethering of an N-terminal half of ATG16L1 disrupted LC3 lipidation in cells in a manner dependent on CCD sequences (Park *et al*, 2016). It remains to be addressed how persistent localisation of ATG16L1 to the PAS can inhibit downstream lipidation of LC3. These inhibitory effects suggest that the displacement of ATG16L1, and potentially upstream autophagy machinery, is required for autophagosome growth. The mechanism through which this occurs remains to be elucidated. It is possible that the overall growth and curvature of the autophagosome may induce the release of autophagy players. Alternatively, it is also possible that additional factors, for instance local changes in calcium levels at the ER (Engedal *et al*, 2013) or post-translational modifications, may aid in neutralising acidic residues located on the lipid-exposed face of the CCD (including D164, E165, E178, E185 or E186). Such regulatory modifications may enhance the electrostatic interactions with the negatively charged lipids and modulate ATG16L1 PAS recruitment. Our data suggest that such fine-tuning of this localisation is essential for proper maturation of autophagosomes.

# Materials and Methods

## Cell culture

Wild-type mouse embryonic fibroblasts (MEFs), ATG3 knockout MEFs (−/−, kind gift from Dr Masaaki Komatsu), ATG16L1$^{-/-}$, ATG5$^{-/-}$ (kind gift from Dr Noboru Mizushima), WIPI2$^{-/-}$, U2OS and 293T cells were cultured in DMEM supplemented with 10% FBS, L-glutamine (2 mM), penicillin (10 units/ml) and streptomycin (0.1 mg/ml). Stable overexpression of GFP-LC3, GFP-ATG or F-S-ATG proteins was obtained by retroviral infection using pBabe expression plasmids followed by blasticidin or puromycin selection. For transient expression, lipofectamine 2000 (Invitrogen) was used according to the manufacturer's instructions. WIPI2, ATG16L1 and Rab33B knockout cells were generated by CRISPR/Cas9-mediated gene editing in wild-type MEFs and transiently expressing Cas9 and gRNA constructs followed by single clone selection for deleted lines. The following gRNA sequences were used to target WIPI2 or ATG16L1: 5′ GCTCTA CATACACAACATC or 5′ AAAGCATGACATGCCAAAT, respectively. For Rab33B knockout, a combination of the following gRNA sequences was used: 5′ GACTTCCGAGAGCGAGCCG, 5′ CGCTCTCCA TCAATATCCA and 5′ ACCGCACCGAGGCCACGAT.

## Antibodies, reagents and treatments

The following antibodies were used: anti-LC3 (Sigma, #L7543); anti-Actin (Sigma, #A5316); anti-p62/SQSTM1 (Cell Signalling, #5114); anti-ATG16L1 (MBL, #PM040); anti-α-tubulin (Calbiochem, #CP06); anti-S tag (Bethyl Laboratories, A190-135A); anti-ATG5 (Sigma, #A0731); anti-FIP200 (Abcam, #ab176816); anti-T7 tag HRP (Novagen); anti-FLAG (Sigma, M2); anti-β-integrin (Cell Signalling, #4706); WIPI2 (Bio-Rad, 2A2); anti-ATG3 (MBL, M133-3); anti-Rab33B (Rab33bd5-Mo-Tk02, Frontier Institute co.); anti-Rabbit (CST, 7074); anti-Mouse (CST, 7076).

Bafilomycin A1 (BafA1, inhibitor of lysosome acidification) was purchased from Sigma or Tocris and used at a final concentration of 20 nM. The following reagents were obtained from Sigma and used at the indicated concentrations: monensin (M5273, 100 μM), ammonium chloride (NH$_4$Cl, A4514, 5 mM), carbonyl cyanide m-chlorophenylhydrazone (CCCP, C2759, 10 or 100 μM) or 3-methyladenine (3′MA, M9281, 5 mM).

For amino acid starvation experiments, cells were grown in DMEM lacking amino acids and serum typically for 2 h prior to harvesting in the presence or absence of BafA1. Glucose starvation was performed by culturing cells in DMEM lacking glucose and sodium pyruvate, supplemented with 10% dialysed FBS and glutamine (2 mM) for 20 h. BafA1 treatment was added 2 h prior to harvesting, as indicated. Control cells were also grown in 10% dialysed FBS and glutamine but in the presence of glucose and sodium pyruvate. For LAP-like assays, cells were treated with monensin, NH$_4$Cl or CCCP typically for 2 h prior to harvesting. For mitophagy assays, cells were treated with CCCP typically for 6 h prior to harvesting.

## Plasmids

Flag-S-tagged (F-S-) mouse ATG16L1 (NM_001205391.1) and mutants were cloned into pBabe-F-S- retroviral vectors. ATG16L1 full-length and truncation fragments (Δ1–Δ4 and ΔWD40) were previously described (Gammoh *et al*, 2013). ATG16L1 mutants were obtained by two-step PCR. Mutant Δ2Δ182-205 was generated using the following forward primer: 5′ AAGGAGCTTGCAGAAGCAGCAATTGTGGATGA GACCTCA. To generated ATG16L1$^{LD}$ (I171D K179D R193D), consecutive mutations of each residue were performed using the following forward primers: I171D 5′ TATGACGCCCTGCAGGACACTTTTA CTGCCCTAGAAGAG; K179D 5′ GAAGAGGACCTGAGGAAAACTA CTGAG; R193D 5′ GAACTGGTCACCGACTGGATGGCTGAG. For ATG16L1$^{LE}$, the following forward primer was used: 5′ AACCA GACCCTGAAGGCTGCGTATGACGCCCTGCAG. The second PCR step of the above mutants was performed using the following common forward and reverse primers: 5′ GCAGCAGTCGACATGTCGTCGGGC CTGCGCGC and 5′ GCAGCACAATTGTCAAGGCTGTGCCCACAGCAC, respectively. The final PCR products were cloned into pBabe-F-S- using SalI and MfeI sites. C-terminal tagged ATG16L1$^{WT}$-GFP was previously described (Gammoh *et al*, 2013). ATG16L1$^{LD}$-GFP was subcloned from pBabe-F-S- vector using XhoI and EcoRI sites within ATG16L1 sequences. ATG16L1$^{ΔWD40}$-GFP was cloned by PCR using MfeI and SalI sites and pBabe-F-S-ATG16L1$^{ΔWD40}$ as a template. For recombinant protein expression, ATG16L1 fragments spanning residues 72–307 were cloned into pET28a plasmid using the pBabe-F-S-ATG16L1$^{WT}$ or pBabe-F-S-ATG16L1$^{LD}$ as templates and the following forward and reverse primers: 5′ GCAGCACAATTGGGACATGAT GGTGCGTGGAAT and 5′ CCGAAGTCGACTCAATCTTTACCAGAAGC AGGATG, respectively. For protein purification from insect cells, wild-type ATG16L1 was cloned by PCR into a pFastBac plasmid containing His-Flag tags and using SalI and XhoI restriction sites and the following forward and reverse primers: 5′ GCAGCAGTCGACATGTC GTCGGGCCTGCGCGC and 5′ GCAGCAGTCGACTCAAGGCTGTGCC CACAGCAC, respectively. GFP-Rab33B was generated by PCR amplification of human Rab33B cDNA (BC036064.1) and cloning into pBabe-GFP vector using SalI and EcoRI restriction sites and the following forward and reverse primers 5′ GCAGCAGTCGACATGGCTGAGGAG ATGGAGTCG and 5′ GCAGCAGAATTCTTAGCACCAGCACGTCAT TGC, respectively.

## Cell lysis, fractionation and Western blotting

For whole cell lysis, cells were washed twice with ice-cold PBS followed by direct scraping in cell lysis buffer (10 mM Tris pH 7.5, 100 mM NaCl, 1 mM EDTA, 1 mM EGTA, 0.1% SDS, 1% Triton X-100, 1 mM β-ME, 0.5% sodium deoxycholate and 10% glycerol) supplemented with protease inhibitor cocktail V (Fisher Scientific UK). Lysates were cleared by spinning at 20,000 *g* for 10 min at 4°C. Cytosolic and membrane fractions were obtained by sequential incubation in a detergent-free buffer (150 mM NaCl, 25 mM HEPES pH 7.5 and 1.5 mM β-ME) supplemented with digitonin and NP-40, respectively. Lysates were analysed by SDS–PAGE and transferred onto nitrocellulose membranes or in the case of LC3 blotting to PVDF membranes (Bio-Rad). Membranes were blocked in 5% milk-TBST for 30 min or overnight followed by immunoblotting with the indicated antibodies. Membranes were developed under UV light using Clarity™ Western ECL substrate (Bio-Rad, 1705061).

## Pull-down assays

Cell lysates were obtained from 293T cells grown in 10-cm plates and transfected with the indicated plasmids by direct lysing in

NP-40 buffer (150 mM NaCl, 25 mM HEPES pH 7.5, 1.5 mM MgCl$_2$, 1 mM EDTA, 1.5 mM β-ME and 0.5% NP-40) supplemented with protease inhibitor cocktail V (Fisher Scientific UK) and proteasome inhibitor MG132 (Sigma). Cell lysates were cleared by spinning at 20,000 $g$ for 10 min at 4°C and incubated with S-protein agarose (Novagen) for 5 h or overnight at 4°C. Beads were then washed 3 times with NP-40 buffer, and bound proteins were analysed by SDS–PAGE and Western blotting.

### Recombinant proteins purification

T7-His-tagged ATG16L1$_{72–307}$ and mutants were expressed from pET28a plasmid in *Escherichia coli* strain BL21 (DE3, Novagen) and grown overnight at 37°C in 50 ml of lysogeny broth (LB) media, containing 50 μg/ml kanamycin. After the initial incubation, the cell culture was diluted into 800 ml LB media with the selection agent and incubated at 37°C until OD600 reached 0.8. To overexpress the protein, host cells were then induced by 0.3 mM IPTG (Sigma) and the culture was maintained for 5 h at 37°C. The cells were centrifuged at 3,830 $g$ for 5 min, supernatant discarded and pellets stored at −80°C. Cell lysis was performed as previously described (Archna & Scrima, 2017). Briefly, cell pellets were resuspended in 20 ml buffer containing 50 mM HEPES pH 7.0, 5% glycerol, 300 mM NaCl, 5 mM-mercaptoethanol (β-ME), 5 mM MgCl$_2$, 5 μg DNase (Roche) and 5 mg lysozyme (Sigma) per litre of culture. Resuspended cells were homogenised using a syringe-based homogenisation method in the presence of 1 mM phenylmethylsulphonyl fluoride (PMSF), and the cell lysate was centrifuged for 30 min at 21,130 $g$. Cell lysate supernatant was applied on an equilibrated 1 ml Ni-NTA resin (Novagen) column. Following loading of the lysate, the column was washed to remove unbounded proteins using 50 ml of buffer (20 mM Tris pH 7.5 300 mM NaCl, 2 mM β-ME, 20 mM imidazole). Protein elution was performed using increasing concentrations of imidazole, ranging from 20 to 250 mM, and collected as 1 ml fractions. Eluted fractions were dialysed for 2 h in buffer containing 150 mM NaCl, 50 mM HEPES pH 7.5 and 1 mM DTT.

Purification of wild-type ATG16L1 from insect cells was performed by the MRC Protein Phosphorylation and Ubiquitylation Unit Reagents and Services, Dundee—UK. Of note, this protein was not used in the liposome co-sedimentation assay as it was prone to pelleting in the absence of liposomes at high centrifugation speeds.

### Microscopy

For fluorescence analyses, cells were grown on glass coverslips in a 6-well plate. 24 h later, cells were either left untreated or treated as indicated. Coverslips were then fixed with 3.7% paraformaldehyde in 20 mM HEPES pH 7.5 for 30 min at room temperature followed by permeabilisation in 0.1% Triton X-100 in PBS for 5 min or cells were fixed and permeabilised by incubating with −20°C methanol on ice for 5 min. Slides were then incubated in primary antibodies in blocking buffer (PBS supplemented with 1% BSA) at 37°C for 2–3 h followed by incubation with Alexa Fluor secondary antibodies (Invitrogen) for 30 min at room temperature. DAPI (Sigma, D9542) was then used to stain nucleic acids. Following extensive washes, coverslips were mounted on microscope slides and images were acquired using a Leica SP5 microscope.

### Liposome preparation and co-sedimentation assay

All lipids were purchased from Avanti unless otherwise indicated. Small unilamellar vesicles (SUVs) were prepared by drying chloroform-dissolved lipids under nitrogen air followed by desiccation under vacuum for 0.5 h. A buffer containing 150 mM NaCl, 50 mM HEPES pH 7.5 and 1 mM DTT was added to rehydrate the dried lipid layer and produce a final lipid concentration of 1 mg/ml. Following 30-min incubation, liposomes were either sonicated in a water bath for 5–10 min or extruded using 100-nm filters and Avanti mini extruder.

For liposome co-sedimentation assays, 50 ng of His-T7-tagged recombinant ATG16L1 was mixed with 30 μl of 1 mg/ml liposomes (35% DOPC, 35% DOPS, 25% DOPE, 5% PI3P) along with BSA to a final concentration 0.5 mg/ml and incubated for 30 min at room temperature. Samples were then transferred to Thickman Polypropylene tubes (Beckman Coulter, 347287) containing liposome binding buffer to yield a final volume of 500 μl and ultracentrifuged for 10 min at 180,000 $g$ and 22°C (Optima Max Beckman Coulter Ultracentrifuge, TLA 120.2 rotor). Equal proportions of the pellet and supernatant were analysed by SDS–PAGE and Western blotting using anti-T7-HRP antibodies.

### Microscopy-based protein–liposome interaction assay

For Fig 3B, Hap1 cells were grown in suspension at 37°C in a 3 l Wheaton spinner flask for 5 days, harvested by centrifugation at 1,300 $g$ for 15 min at 4°C and washed three times with PBS. Pellets were flash-frozen in liquid nitrogen, resuspended in ice-cold liposome binding buffer (150 mM NaCl, 50 mM Tris pH 7.5, 1 mM DTT supplemented with complete protease inhibitors EDTA-free cocktail, Roche Diagnostics) and cleared by centrifugation with 13,000 $g$ at 4°C for 15 min. For purification of ATG16L1, 50 μl of StrepTactin Sepharose High performance beads (GE Healthcare) was added to 4 mg total protein in the supernatant and incubated for 2 h at 4°C. Beads were washed four times with liposome binding buffer. For microscopy-based protein–liposome interaction assay (Fracchiolla *et al*, 2016), 1 μl of StrepTactin beads covered with ATG16L1 was incubated with 15 μl of liposomes (1 mg/ml) prepared by extrusion (100 nm) and contained the following composition 39.5% DOPC, 35% DOPS, 20% DOPE, 5% phosphoinositides (as indicated in the figure) and 0.5% rhodamine-phosphoethanolamine (L1392, Thermo scientific). The binding reaction was incubated for 15 min at room temperature.

For Figs 3E and EV1A, mouse ATG16L1-GFP and mutants were expressed by transient transfection in 293T cells. 24–48 h later, cells were lysed in cell lysis buffer (as above) and subjected to pull down using GFP-Trap beads (Chromotek) for 2 h at 4°C. Beads were then washed three times in lysis buffer followed by three washes in liposome binding buffer (150 mM NaCl, 50 mM Tris pH 7.5, 1 mM DTT). For Fig EV3B, 15 ng of T7-His-tagged ATG16L1$_{72–307}$$^{WT}$ was incubated with Ni-NTA resin for 1 h at 4°C. Beads were then washed 3 times in liposome binding buffer. Beads were then mixed with 1 μl of 1 mg/ml extruded liposomes containing 35% DOPC, 35% DOPS, 23% DOPE, 5% PI3P and 2% rhodamine-DOPE as well as 0.25 mg/ml BSA in a final volume of 40 μl and incubated at room temperature for 10 min.

All samples were imaged immediately by loading onto a 96 glass bottom plate and using a confocal microscope equipped with a 20× objective. Relative liposome binding to beads was quantified in ImageJ by taking maximum brightness along a straight line drawn on the beads and subtracting the values to adjacent empty regions of the image (background fluorescence). Liposome binding of ATG16L1$^{LD}$ in Fig 3E was normalised to the GFP signal of ATG16L1$^{WT}$.

### Lipid beads binding assay

The following lipid beads were purchased from Echelon Biosciences: PI3P beads (P-B003a), PI(3,4)P2 beads (P-B034a), PI (3,4,5)P3 beads (P-B345a), phosphatidylserine beads (PS, P-B0PS), phosphatidic acid beads (PA, P-B0PA), phosphatidylethanolamine beads (PE, P-B0PE) and PIP control beads (P-B000). For pull-down assays, 20 ng of recombinant wild-type ATG16L1 purified from insect cells was incubated with 5 μl beads and 0.1 mg/ml BSA diluted in liposome binding buffer supplemented with 0.5% Igepal. After 1-h incubation at 4°C with rotation, beads were washed three times in Igepal-supplemented liposome binding buffer and bound protein was eluted in SDS loading buffer and analysed by Western blotting.

### Structural modelling

The i-TASSER webserver (Zhang, 2008) was used to thread the sequence of mouse ATG16L1 onto the structure of the *Saccharomyces cerevisiae* Atg16 (PDB code 3A7P). This produced a helical monomer. PyMol (The PyMOL Molecular Graphics System, Version 2.0 Schrödinger, LLC.) was used to create a coiled-coil dimer by aligning to the yeast structure. A NAMD topology of the dimer was generated using the psfgen plugin of VMD 1.9.3 (Humphrey *et al*, 1996). To generate the lipid bilayer model, the Membrane plugin of VMD was used to build a rectangular matrix of PI3P embedded in phosphatidylcholine onto which the dimer was orientated. The VMD script combine.tcl was used to merge the protein and membrane models and remove sterically clashing lipids. It was then solvated and neutralised by the addition of TIP3P water molecules (Jorgensen *et al*, 1983) and Na$^+$ and Cl$^-$ ions (to a concentration of 150 mM) to form a 125.1 Å × 123.7 Å × 87.4 Å simulation box. The full system comprised 123,533 atoms. NAMD 2.12 (Phillips *et al*, 2005) was used for simulating this system, utilising the CHARMM36 force field (Huang *et al*, 2017) in a Langevin temperature and pressure controlled (NPT @ 310K) ensemble with periodic boundary conditions and particle-mesh Ewald electrostatics. Following 10,000 energy minimisation steps (2 fs/step) to remove van der Waals clashes within the system, a production run of 10 ns was performed. The Cα RMSD of residues 164–198 between starting structure and final frame of the simulation was 2.05 Å, indicating that protein association with lipid was stable over this timeframe.

### Ferroptosis assay

Induction of ferroptotic cell death was performed as previously described (Gao *et al*, 2015). Briefly, ATG16L1$^{-/-}$ cells stably expressing F-S-ATG16L1 constructs were seeded in a 12-well dish

for 24 h followed by treatment in DMEM lacking amino acids and supplemented with 10% FBS (to induce ferroptosis) or 10% dialysed FBS (diFBS) as a control. 24 h later, cells were stained with propidium iodide (PI) and imaged using a Leica DM IL LED microscope.

### Graphs and statistical analyses

Graph-making and statistical analyses were performed on Prism 7 (GraphPad). All quantifications were performed on a minimum of three independent experiments. Statistical significance was measured by performing unpaired Student's *t*-tests on no more than two data sets per analysis. Quantifications of immunofluorescence dot numbers and intensities were conducted in ImageJ. For the latter, a suitable threshold was set in ImageJ and the "Analyze particles" feature was used to record the mean grey values of dots. LC3-II/LC3-I ratio densitometry analysis was performed in ImageLab.

**Expanded View** for this article is available online.

### Acknowledgements
We thank Simon Wilkinson for feedback on the manuscript and members of the N.G. laboratory for critical reading of the manuscript and discussions. N.G. is supported by a Cancer Research UK fellowship (C52370/A21586). S.M. is supported by an ERC grant (No. 646653).

### Author contributions
LJD, AGC, ANM, TM and NG performed the experiments and analysed the data. JAM and DRH performed structural modelling and analyses. MS and SM optimised, performed and analysed the microscopy-based protein–liposome interaction assay. XJ provided crucial input. LJD and NG wrote the manuscript.

### Conflict of interest
The authors declare that they have no conflict of interest.

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
