## [Review Process File · The EMBO Journal]

Intrinsic lipid-binding activity of ATG16L1 supports efficient membrane anchoring and autophagy

Leo J Dudley, Ainara G Cabodevilla, Agata N Makar, Martin Sztacho, **Tim Michelberger**, Joseph A Marsh, Douglas R Houston, Sascha Martens, Xuejun Jiang and Noor Gammoh.

Review timeline:

Submission date:	27 th August 2018
Editorial Decision:	9 th October 2018
Revision received:	10 th December 2018
Editorial Decision:	21 st January 2019
Revision received:	4 th February 2019
Accepted:	7 th March 2019

Editor: Elisabetta Argenzio

Transaction Report:

1st Editorial Decision

9th October 2018

Thank you for submitting your manuscript entitled "Intrinsic lipid binding activity of ATG16L1 is essential for membrane anchoring and autophagy" (EMBOJ-2018-100554) to The EMBO Journal. Your study has been sent to three referees for evaluation, and we have now received reports from them, which are enclosed below for your information.

As you can see, while the referees consider the findings interesting, they also raise critical points that need to be addressed before they can support publication in The EMBO Journal. In particular, the referees find that the nature of the ATG16L1 puncta seen in ATG16L1LE mutant cells should be investigated further (points 1 and 4 from referee #1 and #3, respectively). Importantly, we agree with referee #2 that the direct lipid-binding ability of the ATG16L1 CCD domain should be tested by independent biochemical assays and that a role for the WD40 domain of ATG16L1 in lipid binding needs to formally be ruled out. Referee #3 points out that the RAB33B binding activity and the lipid-binding specificity of ATG16L1 need in depth investigation. In addition, and while this was not specifically mentioned by the referees, we had to notice that functional consequences of lipid binding by ATG16L1 in cell are only described up to the level of puncta formation and PAS recruitment. This issue was also mentioned by an external advisor we consulted before sending the manuscript out for peer-review. From our side, and to further strengthen the revised manuscript, we would therefore encourage you to include additional data on the functional contribution from ATG16L1 lipid binding in cells.

Given the referees' interest and overall positive recommendations, I would like to invite you to submit a revised version of the manuscript, addressing the comments of all three reviewers. I should add that it is EMBO Journal policy to allow only a single round of revision, and acceptance of your manuscript will therefore depend on the completeness of your responses in this revised version. In addition, I realize that addressing all the referees' criticisms will require a lot of additional time and effort and be technically challenging. I would therefore understand if you were to choose not to undergo an extensive revision here and rather pursue a submission at an alternative venue, in which case please inform us about your decision at your earliest convenience.

Given the rather open-ended nature of the revision experiments - and to make sure that you focus on experiments that will help you address the most crucial referee concerns - I would like to discuss the data that you could imagine including in a revised manuscript. Could you therefore please provide me with a preliminary point-by-point response to the referee concerns in which you outline your plans for the revision? Based on this, I would be happy to discuss the exact requirements for the revised manuscript with you.

REFEREE REPORTS

Referee #1:

This manuscript entitled "Intrinsic lipid binding activity of ATG16L1 is essential for membrane anchoring and autophagy" by Leo J Dudley et. al. reported the PI3P binding potential in Atg16L1 for the first time. They discovered that conserved residues within the CCD of ATG16L1 that mediate direct binding to PI3P. This binding potential was critical in Atg16L1 recruitment to PAS and autophagy progression. Interestingly, the point mutant that strengthen the binding provided negative effect in autophagy, suggesting proper regulation of PI3P binding is crucial for autophagy. The data are clean and convincing. Suggested model would be mostly consistent with the past literatures, and will provide fundamental insight into autophagy mechanism. One point this referee ask to deepen this study is about the nature of Atg16L1 positive puncta in ATG16L1LE mutant as follows.

Major point

1. What is the nature of Atg16L1 positive puncta in ATG16L1LE mutant? Is it positive for other Atg proteins including FIP200 and WIPI2b? This is important point because if they are positive, Atg16L1 could recruit those proteins and provide novel hierarchy in Atg relationship.

Referee #2:

The present study focuses on a new feature of ATG16L1, an intrinsic lipid-binding ability. A direct binding between ATG16L1 and lipid membrane has not been reported, which makes this subject important for the autophagy field. There is no doubt that the triple mutation (I171D, K179D, and R193D) in the CCD of ATG16L1 is affecting the PAS and autophagy. But that can be expected when a mutation in a relatively rigid α -helix is introduced into a protein. A novel contribution would be represented by linking this in vivo change to a convincing experimental demonstration that the CCD domain of ATG16L1 has a direct lipid-binding ability. In this aspect, the present study falls short, and additional work is needed to address major issues.

1. The microscopy-based technique in Figure 3B and C represents the only in vitro experimental approach to the binding of full-length ATG16L1 to lipids. At least two independent biochemical methods should confirm that. Why didn't the authors use the liposome co-sedimentation assay in Fig. 3?
2. The authors show in Fig.3 A and D and Fig. 6A that the rigid helix in the CCD domain, in between residues L162 and P249, has I171, K179 and R193 on the interface contacting the lipid membrane. However, the same face of the helix also contains E178, E185 and E186. It is difficult to imagine these 3 negatively charged residues not interfering with binding of I171, K179 and R193 to negatively charged phospholipids. In other words, why do E178, E185 and E186 not engage in electrostatic repulsions from the negatively-charged membrane? The authors need to address this issue.
3. The lipid-binding experiment in Fig. 3B, C, E was carried out with full-length ATG16L1-GFP in the presence of small unilamellar vesicles (SUVs). The authors see the strongest binding with SUVs enriched with PI3P. In Fig. 6C, the authors shows a liposome co-sedimentation assay with a much smaller wild-type peptide, ATG16L1[72-307] lacking the WD40 domain. This small peptide does not bind to SUVs at all--there is no protein in the P fraction. How is it that the large GFP-tagged protein binds, but a smaller peptide does not bind to SUVs? The authors need to address this

discrepancy. Of course, the techniques in Fig. 3 and Fig. 6 are different, but they should be consistent with each other.

4. This leads to my most significant point, which is that the authors completely overlooked a possible effect of the WD40 domain. The crystal structure of this domain was published recently (Bajagic et al., *Prot Sci* 2017), and shows that WD40 is a 7-bladed β -propeller. An earlier paper (Baskaran et al., *Mol Cell* 2012) shows that 7-bladed β -propeller structures (e.g. WIPI2, Atg18, etc.) can bind PI3P. The ATG16L1 WD40 domain and WIPI-2 have homology and the WD40 of ATG16L1 has two positively charged pockets. Therefore, it cannot be excluded that the WD40 of ATG16L1 has an intrinsic membrane binding ability. Can the missing WD40 in the wild-type ATG16L1[72-307] peptide be a reason for not seeing this peptide in the P fraction (Fig. 6C)? Could be the preferential binding of ATG16L1-GFP to PI3P-enriched SUVs actually be a result of WD40 binding to lipids (Fig. 3)?

5. ATG16L1 carries a WD40 domain and binds WIPI2. Thereby, it has at its disposal 1-2 membrane binding modules that prefer PI3P. Why would the protein need a third module, the CCD domain, for membrane binding? To convincingly prove the membrane binding ability of the CCD domain in ATG16L1, the authors need to carry out all microscopy-based and liposome-based experiments with ATG16L1 lacking the WD40 domain. If a membrane binding mechanism via the CCD mediates recruitment of the protein to the PAS, the authors need to show this using the triple mutant of ATG16L1 that has the disabled WD40 domain along with the WIPI2 and FIP200 binding sites in ATG5^{-/-} cells.

Minor points:

1. The experiment in Fig. 2E and Fig. 3G needs to be done with disabled WD40, because it has been proposed that the WD40 might facilitate dimerization via a highly conserved interface (Bajagic et al., *Prot Sci* 2017).
2. The representative image in Fig. 3E does not reflect the quantification, a better image should be chosen.

Referee #3:

ATG16L1 is an essential factor for autophagosome formation and together with the ATG12-ATG5 conjugate promotes lipidation of ATG8 family proteins such as LC3. To fulfill its role in autophagy, ATG16L1 must be recruited to preautophagosomal structures (PAS) at an appropriate time; however, its precise targeting mechanism is not fully understood. In this manuscript, Dudley and coworkers showed that a previously less-characterized region (AA185-205) of the coiled-coil domain of ATG16L1 is required for its recruitment to the PAS and that the corresponding region has the ability to bind phosphoinositides such as PI3P. Furthermore, they found that ATG16L1 mutants with decreased or increased PI3P binding activity failed to promote LC3 lipidation. Finally, the authors proposed that a proper lipid binding ability of ATG16L1 is essential for its localization at the PAS as well as LC3 lipidation during starvation-induced autophagy. Overall, the manuscript is well-written and well-organized, and their findings are potentially interesting to the general readers of EMBO J. To strengthen the authors' conclusions, however, followings points need to be addressed prior to publication.

Major points:

1. The authors claimed in the text that sequences within the CCD dispensable for known protein-protein interactions are required for the recruitment of ATG16L1 to the PAS. However, this statement is not accurate. Previous reports have already shown that the C-terminal portion of the CCD binds to RAB33B and that this region is required for starvation-induced autophagy and Salmonella-induced autophagy (MBoC, PMID: 18448665; JCB, PMID: 24100292). Thus, the authors should investigate the RAB33B binding activity of WT and LD/LE mutants of ATG16L1 to exclude the possibility that the LD/LE mutations also affect RAB33B binding.
2. Lipid binding specificity of ATG16L1 and its mutant should be investigated in more detail. Other phospholipids such as PI(3,4)P₂, PI(3,4,5)P₃, PS, and PA should also be investigated in Figures 3

and 6.

3. In Figure 6C, the authors showed the weak interaction between recombinant ATG16L1(AA72-307) and liposomes. However, this reviewer cannot see any significant binding activity, because no difference in the pellet fraction of ATG16L1(AA72-307) was observed in the presence and absence of liposomes. Thus, the most straightforward explanation is that purified ATG16L1(AA72-307) is unable to bind to lipids, and the ATG5-ATG12/ATG16L1 complex rather than ATG16L1 alone mediates lipid binding (see EMBO J., PMID:23064152). Although the liposome binding activity was clearly observed by microscopy-based protein-liposome binding assays (Figure 3B and 3E), the authors cannot exclude the possibility that the ATG16L1 beads used are contaminated by a small amount of endogenous ATG5 during the purification process, because they expressed recombinant ATG16L1 in 293T cells, which endogenously express ATG5. Thus, the authors should use recombinant ATG16L1 from bacteria or ATG5-KO cells to exclude this possibility. Alternatively, the ATG16L1-delta2 mutant lacking an ATG5-binding site could be used for microscopy-based protein-liposome binding assays.

4. In Figure 7D, it is not clear why ATG16L1-LE shows punctate structure even under fed conditions. Is this an autophagy-related structure? IF analysis using phagophore and autophagosome markers should be performed.

5. In Figure 7E, the authors should detect p62 in addition to LC3 to confirm the impact of the LE mutant on autophagy. Furthermore, the expression level of the WT and LE mutant is not comparable (middle panel in Figure 7E). The results should be compared with same/similar expression levels.

Other minor points:

6. For multiple comparisons, t-test should not be used. Appropriate statistical analyses are necessary.

7. For general readers, addition of molecular size markers to each blot is highly recommended in all figures.

Referee #1:

This manuscript entitled "Intrinsic lipid binding activity of ATG16L1 is essential for membrane anchoring and autophagy" by Leo J Dudley et. al. reported the PI3P binding potential in Atg16L1 for the first time. They discovered that conserved residues within the CCD of ATG16L1 that mediate direct binding to PI3P. This binding potential was critical in Atg16L1 recruitment to PAS and autophagy progression. Interestingly, the point mutant that strengthen the binding provided negative effect in autophagy, suggesting proper regulation of PI3P binding is crucial for autophagy. The data are clean and convincing. Suggested model would be mostly consistent with the past literatures, and will provide fundamental insight into autophagy mechanism. One point this referee ask to deepen this study is about the nature of Atg16L1 positive puncta in ATG16L1LE mutant as follows.

Major point

1. What is the nature of Atg16L1 positive puncta in ATG16L1LE mutant? Is it positive for other Atg proteins including FIP200 and WIPI2b? This is important point because if they are positive, Atg16L1 could recruit those proteins and provide novel hierarchy in Atg relationship.

We thank the referee for their positive recommendation. As suggested, we tested the nature of puncta formed by the lipid-enhanced mutant, ATG16L1^{LE}, under basal conditions and show that these structures stain positive for WIPI2 and are responsive to PI3P depletion by 3'MA (newly added Fig EV4A-C). These findings indicate that the puncta formed by ATG16L1^{LE} mutant under basal conditions correspond to pre-autophagosome structures that are likely to be stalled as this mutant cannot support LC3 lipidation.

Referee #2:

The present study focuses on a new feature of ATG16L1, an intrinsic lipid-binding ability. A direct binding between ATG16L1 and lipid membrane has not been reported, which makes this subject important for the autophagy field. There is no doubt that the triple mutation (I171D, K179D, and R193D) in the CCD of ATG16L1 is affecting the PAS and autophagy. But that can be expected when a mutation in a relatively rigid α -helix is introduced into a protein. A novel contribution would be represented by linking this in vivo change to a convincing experimental demonstration that the CCD domain of ATG16L1 has a direct lipid-binding ability. In this aspect, the present study falls short, and additional work is needed to address major issues.

We thank the referee for their positive recommendation.

1. The microscopy-based technique in Figure 3B and C represents the only in vitro experimental approach to the binding of full-length ATG16L1 to lipids. At least two independent biochemical methods should confirm that. Why didn't the authors use the liposome co-sedimentation assay in Fig. 3?

This is indeed an important point. Purified full length ATG16L1 is prone to sedimentation in the absence of liposomes in our system. To overcome this limitation, we used the truncation fragment (residues 72-307) in the liposome co-sedimentation assay which has been previously shown to enhance the solubility of ATG16L1. We added a note to clarify this point on pg.14 of the revised manuscript. In order to confirm the lipid binding activity of wild type ATG16L1 using an independent biochemical assay, we tested the binding of full length ATG16L1 to commercially available beads conjugated to lipids. The newly added Fig EV1C confirms the ability of full length ATG16L1 to bind PI3P, relevant to its localisation to pre-autophagosome structures, as well as to other lipids (discussed below, referee #3 point #2).

2. The authors show in Fig.3 A and D and Fig. 6A that the rigid helix in the CCD domain, in between residues L162 and P249, has I171, K179 and R193 on the interface contacting the lipid membrane. However, the same face of the helix also contains E178, E185 and E186. It is difficult to imagine these 3 negatively charged residues not interfering with binding of I171, K179 and R193 to negatively charged phospholipids. In other words, why do E178, E185 and E186 not engage in electrostatic repulsions from the negatively-charged membrane? The authors need to address this issue.

The referee's suggestion that other acidic residues facing positions 'f' and 'g' on the lipid-interacting interface (including E178, E185 and E186) will affect lipid binding activity of the CCD in a similar manner as mutating residues D164 and E165 is consistent with our model. We have modified the revised manuscript to further emphasise this point on pg.8 of the results section and pg.11 of the discussion.

3. The lipid-binding experiment in Fig. 3B, C, E was carried out with full-length ATG16L1-GFP in the presence of small unilamellar vesicles (SUVs). The authors see the strongest binding with SUVs enriched with PI3P. In Fig. 6C, the authors shows a liposome co-sedimentation assay with a much smaller wild-type peptide, ATG16L1[72-307] lacking the WD40 domain. This small peptide does not bind to SUVs at all--there is no protein in the P fraction. How is it that the large GFP-tagged protein binds, but a smaller peptide does not bind to SUVs? The authors need to address this discrepancy. Of course, the techniques in Fig. 3 and Fig. 6 are different, but they should be consistent with each other.

To address the referee's comment, we used the imaging-based technique to test liposome recruitment to recombinant ATG16L1 fragment (ATG16L1₇₂₋₃₀₇^{WT}) immobilised on beads. The newly added Fig EV3B shows that recombinant ATG16L1₇₂₋₃₀₇^{WT} can positively recruit liposomes similar to what is observed using full length ATG16L1-GFP. This suggests that the discrepancy between the liposome binding techniques (imaging-based and co-sedimentation assays) is likely due to differences in the experimental approach and sensitivity of the readout. We further discuss this on pg.8 of the revised manuscript.

4. This leads to my most significant point, which is that the authors completely overlooked a possible effect of the WD40 domain. The crystal structure of this domain was published recently (Bajagic et al., Prot Sci 2017), and shows that WD40 is a 7-bladed β -propeller. An earlier paper (Baskaran et al., Mol Cell 2012) shows that 7-bladed β -propeller structures (e.g. WIPI2, Atg18, etc.) can bind PI3P. The ATG16L1 WD40 domain and WIPI-2 have homology and the WD40 of ATG16L1 has two positively charged pockets. Therefore, it cannot be excluded that the WD40 of ATG16L1 has an intrinsic membrane binding ability. Can the missing WD40 in the wild-type ATG16L1[72-307] peptide be a reason for not seeing this peptide in the P fraction (Fig. 6C)? Could be the preferential binding of ATG16L1-GFP to PI3P-enriched SUVs actually be a result of WD40 binding to lipids (Fig. 3)?

The referee raises an interesting point. The role of the WD40 domain in the canonical autophagy pathway has been recently published in the EMBO J (Fletcher et al., 2018) showing that it is dispensable for LC3 lipidation during starvation induced autophagy but required for LC3 lipidation on single membranes during LC3-associated phagocytosis. It is also important to note that lipid binding deficient mutant of ATG16L1 (ATG16L1^{LD}) used in Fig 3E harbours an intact WD40 domain but is deficient in recruiting liposomes. Therefore, we predict that the contribution of the WD40 domain to binding lipids in this system is minimal. We directly addressed this point in the revised manuscript using the imaging-based liposome binding assay and show that a truncation mutant of ATG16L1 lacking the WD40 domain is able to bind liposomes in a similar manner to wild type protein (newly added Fig EV1A and B).

5. ATG16L1 carries a WD40 domain and binds WIPI2. Thereby, it has at its disposal 1-2 membrane binding modules that prefer PI3P. Why would the protein need a third module, the CCD domain, for membrane binding? To convincingly prove the membrane binding ability of the CCD domain in ATG16L1, the authors need to carry out all microscopy-based and liposome-based experiments with ATG16L1 lacking the WD40 domain. If a membrane binding mechanism via the CCD mediates recruitment of the protein to the PAS, the authors need to show this using the triple mutant of ATG16L1 that has the disabled WD40 domain along with the WIPI2 and FIP200 binding sites in ATG5^{-/-} cells.

The requirement for multiple modules to target ATG16L1 to membranes is an interesting and open question. It is also interesting that sequences that mediate ATG16L1 binding to WIPI2 and FIP200 and the WD40 domain are absent in yeast Atg16 suggesting that additional conserved mechanisms are likely to exist to target ATG16L1 to the PAS (highlighted in the revised manuscript pg.3). Our results in Fig 4E show that the lipid binding deficient mutant ATG16L1^{LD} is diffused when expressed in cells lacking WIPI2 supporting the conclusion that the interaction of ATG16L1 to lipids and proteins are required for PAS recruitment. As above, ATG16L1 mutant lacking the WD40 domain was able to bind liposomes when tested using the imaging-based assay (newly added Fig EV1A and B). The possibility of the WD40 domain harbouring a lipid binding capacity is however an interesting postulation and is not excluded from our data. Since the WD40 domain is dispensable for canonical autophagy (Fletcher et al., 2018), its postulated lipid binding may have independent functions that are beyond the scope of our manuscript.

Minor points:

1. The experiment in Fig. 2E and Fig. 3G needs to be done with disabled WD40, because it has been proposed that the WD40 might facilitate dimerization via a highly conserved interface (Bajagic et al., Prot Sci 2017).

We tested the dimerisation activity of ATG16L1 lacking WD40 domain in the newly added Fig 2F and show that, unlike an N-terminal truncation mutant lacking CCD sequences (ATG16L1^{Δ3}), ATG16L1^{ΔWD40} retained its ability to dimerise. Importantly, the contribution of the WD40 in ATG16L1 dimerisation is not excluded from our study and the conclusion that lipid binding mutant does not affect the dimerisation of ATG16L1 remains unaltered.

2. The representative image in Fig. 3E does not reflect the quantification, a better image should be chosen.

The representative image in Fig 3E has been replaced in the revised manuscript.

Referee #3:

ATG16L1 is an essential factor for autophagosome formation and together with the ATG12-ATG5 conjugate promotes lipidation of ATG8 family proteins such as LC3. To fulfill its role in autophagy, ATG16L1 must be recruited to preautophagosomal structures (PAS) at an appropriate time; however, its precise targeting mechanism is not fully understood. In this manuscript, Dudley and coworkers showed that a previously less-characterized region (AA185-205) of the coiled-coil domain of ATG16L1 is required for its recruitment to the PAS and that the corresponding region has the ability to bind phosphoinositides such as PI3P. Furthermore, they found that ATG16L1 mutants with decreased or increased PI3P binding activity failed to promote LC3 lipidation. Finally, the authors proposed that a proper lipid binding ability of ATG16L1 is essential for its localization at the PAS as well as LC3 lipidation during starvation-induced autophagy. Overall, the manuscript is well-written and well-organized, and their findings are potentially interesting to the general readers of EMBO J. To strengthen the authors' conclusions, however, followings points need to be addressed prior to publication.

We thank the referee for their positive recommendation.

Major points:

1. The authors claimed in the text that sequences within the CCD dispensable for known protein-protein interactions are required for the recruitment of ATG16L1 to the PAS. However, this statement is not accurate. Previous reports have already shown that the C-terminal portion of the CCD binds to RAB33B and that this region is required for starvation-induced autophagy and Salmonella-induced autophagy (MBoC, PMID: 18448665; JCB, PMID: 24100292). Thus, the authors should investigate the RAB33B binding activity of WT and LD/LE mutants of ATG16L1 to exclude the possibility that the LD/LE mutations also affect RAB33B binding.

As suggested by the referee, residues within the middle region of ATG16L1 (W194 and M195) have been shown to be required for autophagy when mutated in conjunction with additional sequences (combined mutations in residues 239-242 and deletion of the WD40 domain) (Fujita et al., 2013). It is important to note that these residues do not overlap with those required for lipid binding identified in our study (I171, K179 and R193). Furthermore, the middle region of ATG16L1 was also shown to bind Rab33B. However, genetic knockdown of Rab33B did not affect LC3 lipidation while overexpression of a constitutively active mutant of Rab33B or OATL, a RabGAP for Rab33B, abrogated autophagosome formation at a later maturation step (Itoh et al., 2008; Itoh et al., 2011). Collectively, these data suggest that published activities of the CCD and lipid binding identified in our study independently mediate the role of ATG16L1 during autophagy. We have modified the manuscript text to specify that the CCD lipid binding region is dispensable for protein interactions tested in this study (e.g. with FIP200 and WIPI2, changes highlighted throughout the text).

As suggested by the referee, we tested the ability of lipid binding mutants of ATG16L1 to interact with Rab33B using transient expression system in 293T cells. Unexpectedly, we observed that the lipid binding deficient mutant ATG16L1^{LD} exhibited a strongly diminished affinity to Rab33B (newly added Fig EV2A). On the contrary, the lipid-enhanced mutant ATG16L1^{LE} retained its ability to bind Rab33B (newly added Fig EV3A). We further tested whether genetic inhibition of Rab33B implicated autophagy in a similar manner as mutating lipid binding residues in ATG16L1 or inhibiting PI3P synthesis by employing CRISPR/Cas9-mediated deletion of Rab33B. Our findings, included as newly added Fig EV2B-C, show that the absence of Rab33B did not affect LC3 lipidation or ATG16L1 puncta formation in a similar manner to mutating lipid binding residues in ATG16L1 or depleting PI3P by 3'MA treatment. Altogether, these findings suggest that binding of ATG16L1 to Rab33B and lipids result in functionally distinct consequences in cells and may require proper localisation of ATG16L1 or occur in a mutually exclusive manner. The revised manuscript was modified to include a discussion of the relevance of ATG16L1 binding to Rab33B and lipids (pg.10-11).

Note: We aimed to test whether ATG16L1 binding to lipids and Rab33B occurred in a mutually exclusive manner. Unfortunately, we could not detect an interaction between recombinant Rab33B and ATG16L1 in our system therefore we could not perform in vitro competition assays. Future work will involve further characterisation of ATG16L1 binding to its partners which falls beyond the scope of this manuscript.

2. Lipid binding specificity of ATG16L1 and its mutant should be investigated in more detail. Other phospholipids such as PI(3,4)P2, PI(3,4,5)P3, PS, and PA should also be investigated in Figures 3 and 6.

In our system, we observe that ATG16L1 cannot bind liposome preparations lacking PI3P, PI4P or PI(4,5)P2 but containing PS, PE and PC. This suggests that phospholipids, such as PS, PE and PC, are not sufficient to bind ATG16L1 when tested using the liposome co-sedimentation and imaging-based assays. We chose to focus on binding to phosphoinositides as they are associated with membrane compartments where ATG16L1 has been shown to localise and emphasised on PI3P binding given its relevance in the PAS recruitment of ATG proteins.

Using an alternative assay to test lipid binding of ATG16L1 (as suggested by referee #2, point #1), we observed consistent ability of ATG16L1 to bind PI3P and PI(3,4)P2 but not to PE or PS (in agreement with our liposome-based assays, newly added Fig EV1C). This assay also revealed a potential ability of ATG16L1 to bind PA, although the data were not statistically significant. It is possible that the binding of ATG16L1 to other phospholipids, such as PA, may implicate its activities independently of PAS recruitment. Such possibilities are further discussed in the revised manuscript (pg.10).

3. In Figure 6C, the authors showed the weak interaction between recombinant ATG16L1(AA72-307) and liposomes. However, this reviewer cannot see any significant binding activity, because no difference in the pellet fraction of

ATG16L1(AA72-307) was observed in the presence and absence of liposomes. Thus, the most straightforward explanation is that purified ATG16L1(AA72-307) is unable to bind to lipids, and the ATG5-ATG12/ATG16L1 complex rather than ATG16L1 alone mediates lipid binding (see EMBO J., PMID:23064152). Although the liposome binding activity was clearly observed by microscopy-based protein-liposome binding assays (Figure 3B and 3E), the authors cannot exclude the possibility that the ATG16L1 beads used are contaminated by a small amount of endogenous ATG5 during the purification process, because they expressed recombinant ATG16L1 in 293T cells, which endogenously express ATG5. Thus, the authors should use recombinant ATG16L1 from bacteria or ATG5-KO cells to exclude this possibility. Alternatively, the ATG16L1-delta2 mutant lacking an ATG5-binding site could be used for microscopy-based protein-liposome binding assays.

As suggested by this referee and to exclude the contribution of residual ATG5 binding, we purified ATG16L1 from ATG5-/- cells and show that it can bind liposomes in the imaging-based assay (newly added Fig EV1A and B). We also show that the recombinant ATG16L1⁷²⁻³⁰⁷ fragment can bind to liposomes in the imaging-based assay (newly added Fig EV3B). Together, our data suggest that the differences between liposome binding in the co-sedimentation and imaging-based assays are likely due to differences in the experimental systems and readout rather than ATG5 binding (further discussed on pg.8 of the revised manuscript).

4. In Figure 7D, it is not clear why ATG16L1-LE shows punctate structure even under fed conditions. Is this an autophagy-related structure? IF analysis using phagophore and autophagosome markers should be performed.

As suggested by this referee and referee #1, we addressed this issue by staining for ATG16L1^{LE}-positive puncta under fed conditions with endogenous WIPI2 and during 3'MA treatment to deplete PI3P. Our findings, included in the newly added Fig EV4A-C, confirm that ATG16L1^{LE} localises to autophagy-related structures under unfed conditions which are likely to be stalled as this mutant cannot support LC3 lipidation.

5. In Figure 7E, the authors should detect p62 in addition to LC3 to confirm the impact of the LE mutant on autophagy. Furthermore, the expression level of the WT and LE mutant is not comparable (middle panel in Figure 7E). The results should be compared with same/similar expression levels.

We agree with the referee that mutant ATG16L1^{LE} appears to express at lower levels in ATG16L1-deficient background. We addressed this issue by improving our virus titre in order to obtain comparable levels between ATG16L1^{WT} and ATG16L1^{LE}-expressing cells. The updated Fig 7E shows that LC3 lipidation and p62 degradation are strongly inhibited in the newly generated ATG16L1^{LE}-expressing cells further supporting our conclusion that enhanced lipid binding of this mutant suppresses autophagy.

Other minor points:

6. For multiple comparisons, t-test should not be used. Appropriate statistical analyses are necessary.

We have double checked the use of t-tests in our statistical analyses to ensure that only two data sets are being compared. This has been clarified in the figure legends as well as the materials and methods section of the revised manuscript.

7. For general readers, addition of molecular size markers to each blot is highly recommended in all figures.

We thank the referee for this useful suggestion and have included molecular weight markers to all western blots in the revised manuscript.

References:

Fletcher, K., Ulferts, R., Jacquin, E., Veith, T., Gammoh, N., Arasteh, J.M., Mayer, U., Carding, S.R., Wileman, T., Beale, R., *et al.* (2018). The WD40 domain of ATG16L1 is required for its non-canonical role in lipidation of LC3 at single membranes. *EMBO J.*

Fujita, N., Morita, E., Itoh, T., Tanaka, A., Nakaoka, M., Osada, Y., Umemoto, T., Saitoh, T., Nakatogawa, H., Kobayashi, S., *et al.* (2013). Recruitment of the autophagic machinery to endosomes during infection is mediated by ubiquitin. *J Cell Biol* **203**, 115-128.

Gao, M., Monian, P., Pan, Q., Zhang, W., Xiang, J., and Jiang, X. (2016). Ferroptosis is an autophagic cell death process. *Cell Res* **26**, 1021-1032.

Gao, M., Monian, P., Quadri, N., Ramasamy, R., and Jiang, X. (2015). Glutaminolysis and Transferrin Regulate Ferroptosis. *Mol Cell* **59**, 298-308.

Itoh, T., Fujita, N., Kanno, E., Yamamoto, A., Yoshimori, T., and Fukuda, M. (2008). Golgi-resident small GTPase Rab33B interacts with Atg16L and modulates autophagosome formation. *Mol Biol Cell* **19**, 2916-2925.

Itoh, T., Kanno, E., Uemura, T., Waguri, S., and Fukuda, M. (2011). OATL1, a novel autophagosome-resident Rab33B-GAP, regulates autophagosomal maturation. *J Cell Biol* **192**, 839-853.

Thank you for submitting a revised version of your manuscript and please accept my apologies for the delay in getting back to you with our decision due to the recent seasonal holidays. Your study has now been seen by the original referees, whose comments are shown below.

As you will see, while referee #1 finds that all criticisms have been sufficiently addressed, referee #2 stresses that your structural model in Figure 3D does not reflect the findings of this work. In particular, this referee requests you to clearly state whether or not you are reporting the first direct electrostatic binding between solvent-exposed residues of the dimeric CCD of ATG16L1 and PI3P or PI(3,4)P2. If not, s/he invites you to discuss in depth any existing example in the literature. Finally, referee #3 recommend you to tone down the title of the manuscript, as the possibility that unidentified binding partners other than lipids are involved in the ATG16L1 recruitment to PAS cannot be excluded at this stage.

REFeree REPORTS

Referee #1:

The authors addressed adequately to my concern, and now the paper is acceptable.

Referee #2:

The revised manuscript provides much better elaboration on binding of the ATG16L1 CCD to lipids. It shows that the ATG16L1 CCD does not bind to phospholipids, such as PI, PS, PE, PC, and PA. The CCD in its dimeric form directly binds only to PI3P and PI(3,4)P2, and the binding residues are solvent-exposed at the positions "f" and "g". I have one major point that I think the authors need to address prior to publication. I think that their structural model in Figure 3D does not reflect the finding of this work. The CCD dimer does not interact directly with lipid bilayer and does not lay flat directly on membrane surface. The authors cannot refer to the studies of Klocek et al. (2009), Pluhackova et al. (2015), and Woo and Lee (2016), because these studies explore very different mechanisms of interaction of a peptide with a lipid bilayer.

In the publication by Klocek et al. (2009), melittin is a random coil peptide that folds into an α -helix upon binding to lipid bilayer, which involves four steps: i) electrostatic attraction of cationic peptide to an anionic membrane surface, ii) hydrophobic insertion into the lipid membrane, iii) a conformational change from random coil to α -helix, and iv) peptide aggregation in the lipid phase. Thus, the random coil melittin is completely unrelated to the well-folded and dimeric CCD of ATG16L1.

Pluhackova et al. (2015) is a study on the adsorption of the model SNARE (E and K) peptides to the phospholipid/cholesterol membrane, where initial binding of the positively charged N terminus of the peptide to the phospholipid headgroups is followed by insertion of the peptide at the interface between the hydrophobic core and polar headgroup region of the membrane. The main interacting force is hydrophobic and involves residues at the positions "a" and "d". Moreover, K and E peptides dissociate from a heterodimeric coiled-coil, bind as single helix, not a dimer, and use mainly hydrophobic residues for interaction with the membrane. In contrast, the residues at the positions "a" and "d" in the CCD of ATG16L1 are hidden in the dimer interface, and are not available for interaction with lipids. The ATG16L1 CCD utilizes solvent-exposed residues at the positions "f" and "g", as clearly suggested by the authors.

Woo and Lee (2016) studied insertions of E and K peptides into lipid bilayers and how the longer hydrocarbons of Lys residues, as compared to Glu residues, form stronger hydrophobic interactions with lipid tails. In contrast, the CCD of ATG16L1 does not interact with the head groups of PI, PE, PS, or PC and thus, its Lys residues cannot reach the lipid tails in the hydrophobic portion of the lipid bilayer.

Direct electrostatic binding between solvent-exposed residues of the dimeric CCD of AT16L1 and PI3P or PI(3,4)P2 proposed by this work is surprising. If this mechanism has been reported for some other dimeric coiled-coil peptide, the authors need to present a correct example in the Discussion section. If no analogous mechanism has been reported, the authors need to clearly acknowledge this, and not mislead or confuse readers with incorrect citations [Klocek et al. (2009), Pluhackova et al. (2015), and Woo and Lee (2016)] of studies that explore completely different structural models. Along this line, a modification of the structural model in Figure 3D is highly recommended.

The finding on the interaction between the dimeric CCD of ATG16L1 and phosphoinositide phosphates will be a major part of the publication. Therefore, a correct structural model is important for good understanding by the general readers of EMBO J.

Referee #3:

In the revised manuscript, the authors performed additional experiments and addressed most of the concerns raised by this reviewer. I have only one minor suggestion. The title "Intrinsic lipid binding activity of ATG16L1 is essential..." is too strong, because the authors cannot exclude the possibility that unidentified binding partners other than lipids are involved in the ATG16L1 recruitment to PAS as described in the Discussion. Thus, I strongly recommend the authors to weaken the expression of the title.

2nd Revision - authors' response

4th February 2019

Referee #1:

The authors addressed adequately to my concern, and now the paper is acceptable.

Referee #2:

The revised manuscript provides much better elaboration on binding of the ATG16L1 CCD to lipids. It shows that the ATG16L1 CCD does not bind to phospholipids, such as PI, PS, PE, PC, and PA. The CCD in its dimeric form directly binds only to PI3P and PI(3,4)P2, and the binding residues are solvent-exposed at the positions "f" and "g". I have one major point that I think the authors need to address prior to publication. I think that their structural model in Figure 3D does not reflect the finding of this work. The CCD dimer does not interact directly with lipid bilayer and does not lay flat directly on membrane surface. The authors cannot refer to the studies of Klocek et al. (2009), Pluhackova et al. (2015), and Woo and Lee (2016), because these studies explore very different mechanisms of interaction of a peptide with a lipid bilayer.

In the publication by Klocek et al. (2009), melittin is a random coil peptide that folds into an α -helix upon binding to lipid bilayer, which involves four steps: i) electrostatic attraction of cationic peptide to an anionic membrane surface, ii) hydrophobic insertion into the lipid membrane, iii) a conformational change from random coil to α -helix, and iv) peptide aggregation in the lipid phase. Thus, the random coil melittin is completely unrelated to the well-folded and dimeric CCD of ATG16L1.

Pluhackova et al. (2015) is a study on the adsorption of the model SNARE (E and K) peptides to the phospholipid/cholesterol membrane, where initial binding of the positively charged N terminus of the peptide to the phospholipid headgroups is followed by insertion of the peptide at the interface between the hydrophobic core and polar headgroup region of the membrane. The main interacting force is hydrophobic and involves residues at the positions "a" and "d". Moreover, K and E peptides dissociate from a heterodimeric coiled-coil, bind as single helix, not a dimer, and use mainly hydrophobic residues for interaction with the membrane. In contrast, the residues at the positions "a" and "d" in the CCD of ATG16L1 are hidden in the dimer interface, and are not available for interaction with lipids. The ATG16L1 CCD utilizes solvent-exposed residues at the positions "f" and "g", as clearly suggested by the authors.

Woo and Lee (2016) studied insertions of E and K peptides into lipid bilayers and how the longer hydrocarbons of Lys residues, as compared to Glu residues, form stronger hydrophobic interactions with lipid tails. In contrast, the CCD of ATG16L1 does not interact with the head groups of PI, PE, PS, or PC and thus, its Lys residues cannot reach the lipid tails in the hydrophobic portion of the lipid bilayer.

Direct electrostatic binding between solvent-exposed residues of the dimeric CCD of ATG16L1 and PI3P or PI(3,4)P2 proposed by this work is surprising. If this mechanism has been reported for some other dimeric coiled-coil peptide, the authors need to present a correct example in the Discussion section. If no analogous mechanism has been reported, the authors need to clearly acknowledge this, and not mislead or confuse readers with incorrect citations [Klocek et al. (2009), Pluhackova et al. (2015), and Woo and Lee (2016)] of studies that explore completely different structural models. Along this line, a modification of the structural model in Figure 3D is highly recommended.

The finding on the interaction between the dimeric CCD of ATG16L1 and phosphoinositide phosphates will be a major part of the publication. Therefore, a correct structural model is important for good understanding by the general readers of EMBO J.

We have modified the manuscript to clarify our conclusions from the structural prediction of ATG16L1-lipid interaction. We rephrased the text to clarify that residues within the CCD are likely to mediate the interaction with phosphorylated phosphoinositide head groups that are embedded within the lipid bilayer rather than mediate the binding to the lipid bilayer directly. We have cited published work that show interactions between CCD's and phospholipids but clarified that the structural bases underlying these interactions have not been directly addressed. The reference Klocek et al. (2009) has been deleted as we agree that this system is not comparable to our proposed model. On the other hand, we have clarified that the references Pluhackova et al. (2015) and Woo and Lee (2016) are also substantially different with the main similarity being that the helices lie flat on the membrane. We thank the reviewer for pointing out this ambiguity in the text. The modified manuscript includes edits in the text (pages 6, 16 and 23) and Figure 3D.

Referee #3:

In the revised manuscript, the authors performed additional experiments and addressed most of the concerns raised by this reviewer. I have only one minor suggestion. The title "Intrinsic lipid binding activity of ATG16L1 is essential..." is too strong, because the authors cannot exclude the possibility that unidentified binding partners other than lipids are involved in the ATG16L1 recruitment to PAS as described in the Discussion. Thus, I strongly recommend the authors to weaken the expression of the title.

We have modified the title to the following: "Intrinsic lipid binding activity of ATG16L1 supports efficient membrane anchoring and autophagy".

Accepted

7th March 2019

I am pleased to inform you that your manuscript has been accepted for publication in the EMBO Journal.

Corresponding Author Name: Noor Gammoh

Manuscript Number: EMBOJ-2018-100554